# The DNA-binding protein HTa from *Thermoplasma acidophilum* is an archaeal histone analog

Antoine Hocher[1,2]*, Maria Rojec[1,2], Jacob B Swadling[1,2], Alexander Esin[1,2], Tobias Warnecke[1,2]*

[1]MRC London Institute of Medical Sciences (LMS), London, United Kingdom; [2]Institute of Clinical Sciences (ICS), Faculty of Medicine, Imperial College, London, United Kingdom

**Abstract** Histones are a principal constituent of chromatin in eukaryotes and fundamental to our understanding of eukaryotic gene regulation. In archaea, histones are widespread but not universal: several lineages have lost histone genes. What prompted or facilitated these losses and how archaea without histones organize their chromatin remains largely unknown. Here, we elucidate primary chromatin architecture in an archaeon without histones, *Thermoplasma acidophilum,* which harbors a HU family protein (HTa) that protects part of the genome from micrococcal nuclease digestion. Charting HTa-based chromatin architecture in vitro, in vivo and in an HTa-expressing *E. coli* strain, we present evidence that HTa is an archaeal histone analog. HTa preferentially binds to GC-rich sequences, exhibits invariant positioning throughout the growth cycle, and shows archaeal histone-like oligomerization behavior. Our results suggest that HTa, a DNA-binding protein of bacterial origin, has converged onto an architectural role filled by histones in other archaea.

**\*For correspondence:**
a.hocher@lms.mrc.ac.uk (AH);
tobias.warnecke@imperial.ac.uk
(TW)

**Competing interests:** The authors declare that no competing interests exist.

## Introduction

Across all domains of life, DNA is intimately associated with proteins that wrap, package, and protect it. Bacteria typically encode multiple small basic proteins that are dynamically expressed and fulfill a variety of architectural roles by bridging, wrapping, or bending DNA (*Dillon and Dorman, 2010*). Some of these proteins are phylogenetically widespread, others restricted to specific lineages (*Swiercz et al., 2013*; *Lagomarsino et al., 2015*). Bacterial chromatin, on the whole, is diverse and dynamic over both physiological and evolutionary timescales. In contrast, a single group of proteins has come to dominate chromatin in eukaryotes: histones. Assembling into octameric complexes that wrap ~ 147 bp of DNA, eukaryotic histones not only mediate genome compaction but also establish a basal landscape of differential accessibility, elaborated via a plethora of post-translational modifications, that is fundamental to our understanding of eukaryotic gene regulation (*Jiang and Pugh, 2009*; *Bai and Morozov, 2010*).

Histones are also widespread in archaea (*Adam et al., 2017*; *Henneman et al., 2018*). They have the same core fold (*Decanniere et al., 2000*; *Mattiroli et al., 2017*) as eukaryotic histones, but lack N- and typically also C-terminal tails, the principal substrates for post-translational modifications in eukaryotes (*Henneman et al., 2018*). Dimers in solution, they assemble into tetramers that wrap ~ 60 bp of DNA (*Bailey et al., 1999*). This minimal nucleosomal unit can be extended, at least in some archaea, into a longer oligomer via incorporation of additional dimers (*Xie and Reeve, 2004*; *Maruyama et al., 2013*; *Mattiroli et al., 2017*). Like their eukaryotic counterparts, archaeal nucleosomes preferentially bind sequences that facilitate wrapping, for example by means of periodically spaced GC/GG/AA/TT dinucleotides (*Bailey and Reeve, 1999*; *Bailey et al., 2000*). On average, nucleosome occupancy is higher on more GC-rich sequences and lower around transcriptional

start and end sites (*Ammar et al., 2011*; *Nalabothula et al., 2013*), which tend to be AT-rich. The precise role of archaeal histones in transcription regulation, however, remains poorly understood (*Gehring et al., 2016*).

Although widespread, archaeal histones are less entrenched than histones in eukaryotes: they have been deleted experimentally in several species without dramatic effects on transcription and growth (*Heinicke et al., 2004*; *Weidenbach et al., 2008*; *Dulmage et al., 2015*) and lost altogether from a handful of archaeal lineages (*Adam et al., 2017*). A particularly intriguing case concerns the thermophilic acidophile *Thermoplasma acidophilum,* which lacks histone genes but instead encodes a protein named HTa (**H**istone-like protein of **T**hermoplasma **a**cidophilum). Based on its primary sequence (*DeLange et al., 1981*; *Drlica and Rouviere-Yaniv, 1987*) and predicted secondary, tertiary and quaternary structure (*Figure 1a–b*), HTa is a member of the HU family of proteins. As in *E. coli*, where the average cell during exponential growth contains an estimated 30,000–55,000 HU molecules (*Figure 1c*; *Ali Azam et al., 1999*), HU proteins are often abundant constituents of bacterial chromatin. They are also frequently essential (*Grove, 2011*) and broadly distributed across bacterial phyla (*Supplementary file 1*). Although individual members of the HU family have diverged in their DNA binding properties, even distant homologs display functional similarities, constraining negative supercoils and binding not only B-form but also structurally unusual DNA such as cruciforms (*Grove, 2011*). Outside bacteria, HU proteins are comparatively rare (*Figure 2a*). They are found with a modicum of phylogenetic persistence only in some single-celled eukaryotes, where they are known to play important functional roles (*Sasaki et al., 2009*; *Gornik et al., 2012*), and in a single clade of archaea: the

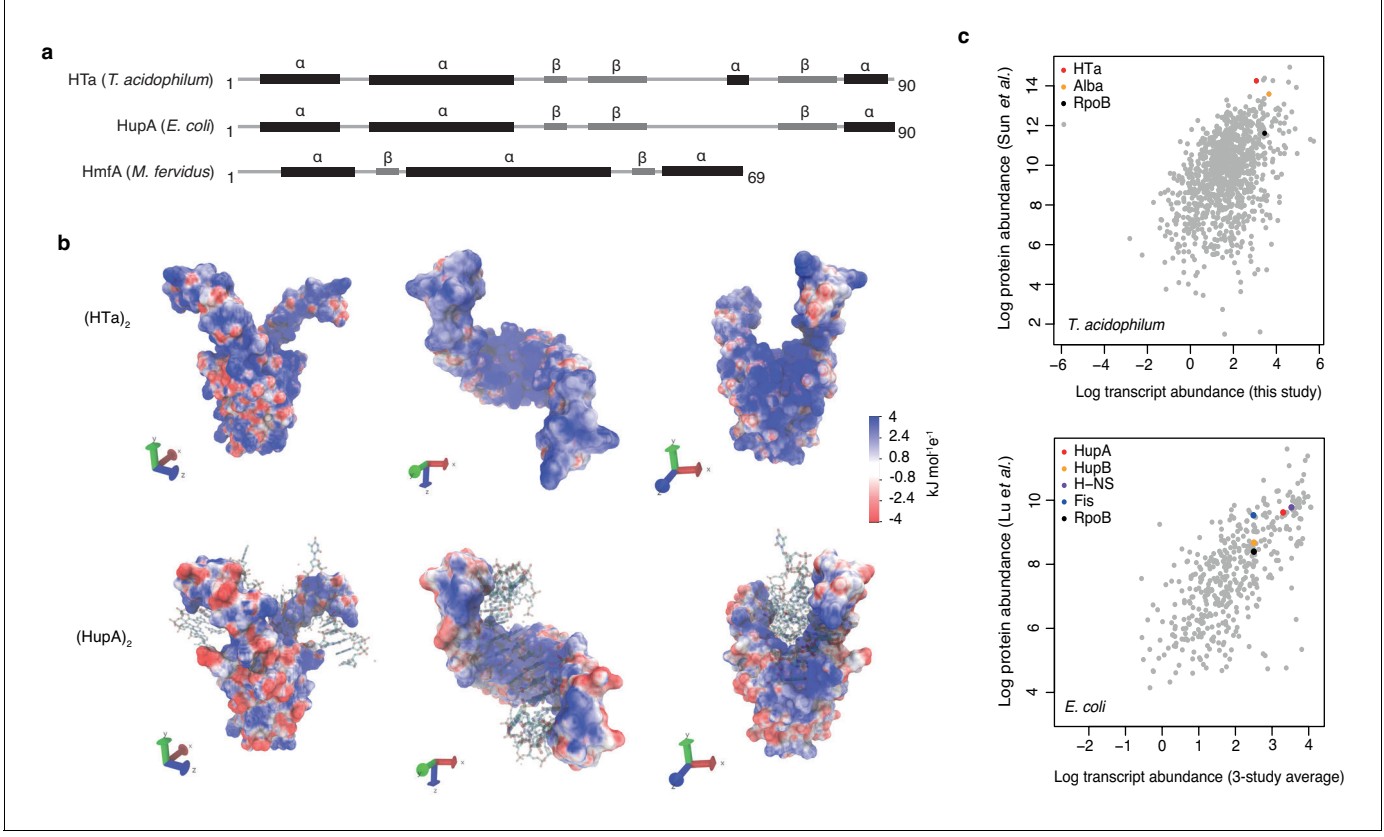

**Figure 1.** Predicted structure and measured abundance of HTa. (**a**) Predicted secondary structures of HTa (*T. acidophilum*), the bacterial HU protein HupA (*E. coli*), and the archaeal histone protein HmfA (*M. fervidus*). (**b**) Predicted quaternary structure of the (HTa)₂ homodimer compared to the crystal structure of (HupA)₂ (PDB: 1p51) bound to DNA. Color gradients represent charge densities mapped onto the solvent accessible surface area of (HTa)₂ and (HupA)₂. Note the extended patches of stronger positive charge for (HTa)₂ compared to (HupA)₂, particularly in the stalk region. (**c**) Correlation of transcript and protein abundances for *T. acidophilum* and *E. coli*. HTa and HU are highlighted along with some additional chromatin-associated proteins. Data sources: *T. acidophilum* protein abundance: ***Sun et al. (2010)***; *E. coli* protein abundance: ***Lu et al. (2007)***. *E. coli* transcript abundance is an average across three previous studies as reported by ***Lu et al. (2007)***.

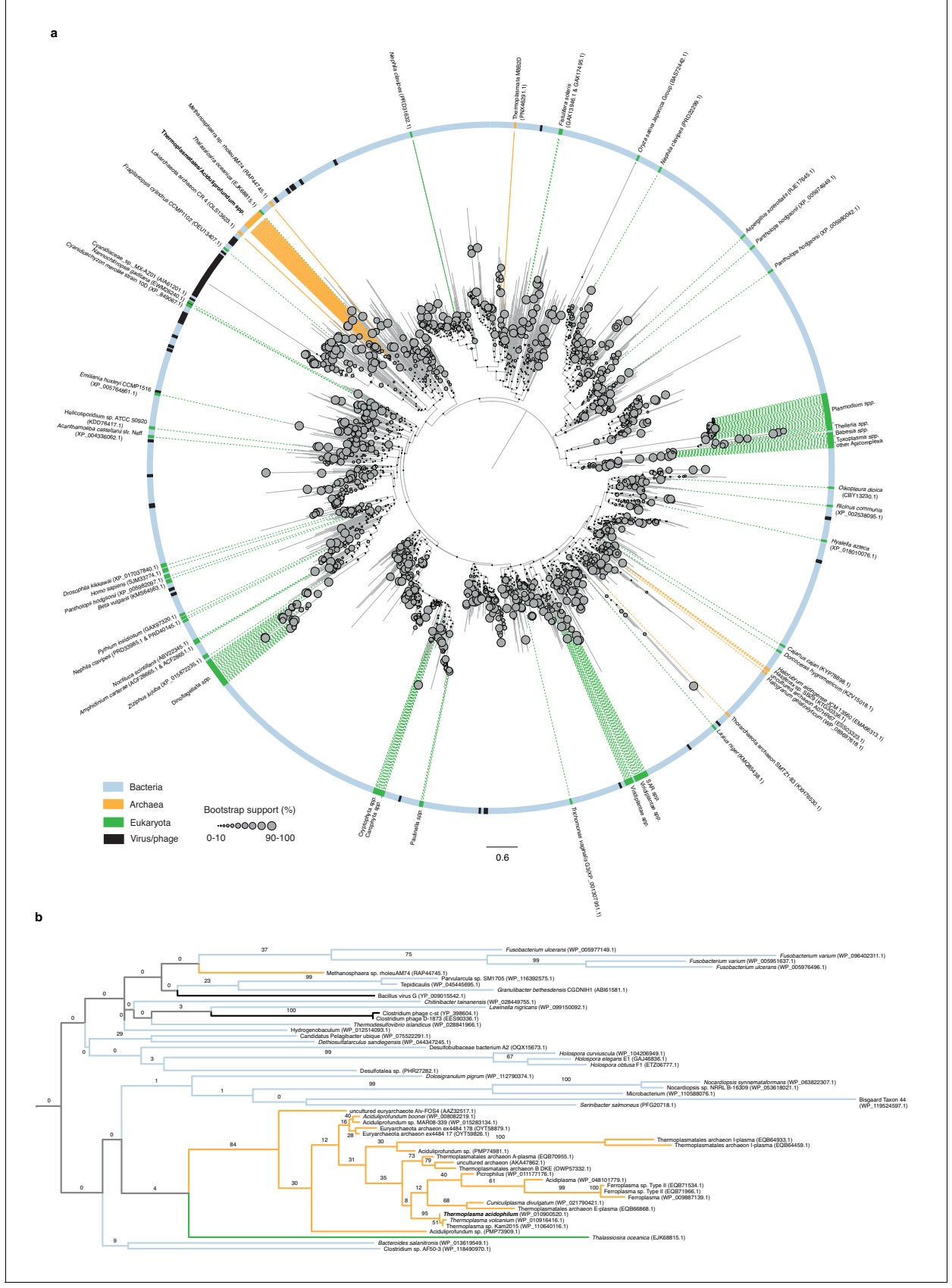

**Figure 2.** Phylogenetic relationships of HU family proteins from bacteria, eukaryotes, and archaea. (a) Protein-level phylogenetic tree of HU proteins including HTa (see Materials and methods for details on phylogenetic reconstruction). The tree is midpoint-rooted. Reported domain-level membership (Bacteria, Archaea, etc.) of different proteins is color-coded in the outer circle and on the dotted lines that point to individual branches. See main text and Materials and methods for a critical evaluation of domain assignments and likely assembly contaminants. Bootstrap support values (%) for individual branches, visually encoded as node diameters, illustrate poorly resolved relationships at deeper nodes. (b) Excerpt of the phylogeny shown above, highlighting good support (84%) for a monophyletic origin of HU proteins in the Thermoplasmatales/DHVE2 clade and their uncertain affiliation to other HU family members.

The online version of this article includes the following figure supplement(s) for figure 2:

**Figure supplement 1.** Phylogenetic placement of HU proteins attributed to halophilic archaea.

Thermoplasmatales/deep-sea hydrothermal vent euryarchaeota (DHVE2 group). Phylogenetic reconstruction of the HTa/HU gene family suggests that HTa was acquired via horizontal gene transfer from bacteria at the root of this clade (*Figure 2b*, see Materials and methods).

In *T. acidophilum*, HTa is highly abundant (*Figure 1c*) and protects ~ 25–35% of the genome from micrococcal nuclease (MNase) digestion (*Searcy and Stein, 1980*; *Thomm et al., 1982*), consistent with a global role in structuring *T. acidophilum* chromatin. Pioneering work by Searcy and colleagues showed that MNase digestion of native *T. acidophilum* chromatin yields two distinct fragment sizes of ~ 40 bp and ~ 80 bp (*Searcy and Stein, 1980*). The same authors observed a similar banding pattern when they digested calf thymus DNA following in vitro reconstitution with purified HTa, suggesting that native HTa is sufficient for and likely the principal mediator of protection from MNase digestion in *T. acidophilum* (*Searcy and Stein, 1980*). It remains unknown, however, where HTa binds to the *T. acidophilum* genome in vivo; whether HTa binds in a sequence-specific manner or promiscuously; whether it requires particular post-translational modifications to carry out its functions; whether binding is dynamic in response to environmental changes; and how binding relates to functional genomic landmarks. Crucially, we do not know how HTa-mediated chromatin organization in *T. acidophilum* compares to that in histone-encoding archaea: do HTa and histones fill similar functional and architectural niches? Or are their binding patterns and functional repercussions entirely distinct?

Here, to begin to address these questions, we characterize genome-wide chromatin organization in *T. acidophilum*. Using MNase treatment coupled to high-throughput sequencing, we find footprints of protection throughout the genome. Confirming prior results (*Searcy and Stein, 1980*), we observe a bimodal distribution of protected fragment sizes and subsequently infer small and large binding footprints. The more common smaller footprints are well predicted by simple sequence features and display a general preference for GC-rich sequences. We observe the same preferences in an HTa-expressing *E. coli* strain, when we reconstitute chromatin using purified HTa and *T. acidophilum* genomic DNA, and in electrophoretic mobility shift assays (EMSAs) of GC-variable DNA oligos, consistent with sequence as a key determinant of HTa binding in vivo. HTa sequence preferences, positioning around transcription start sites, and static behavior throughout the growth cycle are reminiscent of archaeal histones rather than well-characterized bacterial HU homologs. In addition, we present evidence that larger fragments are frequently derived from nucleation-extension events akin to oligomerization of archaeal histones (*Maruyama et al., 2013*; *Nalabothula et al., 2013*; *Mattiroli et al., 2017*). Our results suggest that, in key aspects of its molecular behavior, HTa can be regarded as an archaeal histone analog.

## Results

### Primary chromatin structure across the *T. acidophilum* growth cycle

To elucidate genome-wide HTa binding in vivo, we carried out a series of MNase experiments across the *T. acidophilum* growth cycle (*Figure 3a*). Throughout, MNase digestion yielded protected fragments of two distinct sizes (*Figure 3b*), in line with previous results (*Searcy and Stein, 1980*). MNase digestion of *E. coli* cells expressing recombinant HTa produced a very similar protection pattern (*Figure 3c*). This demonstrates that HTa readily binds DNA outside its native cellular environment and does not require *T. acidophilum*-specific post-translational modifications or binding partners to protect from MNase digestion, consistent with previous in vitro reconstitution experiments (*Searcy and Stein, 1980*). In contrast, native *E. coli* HU (HupA), even when strongly over-expressed

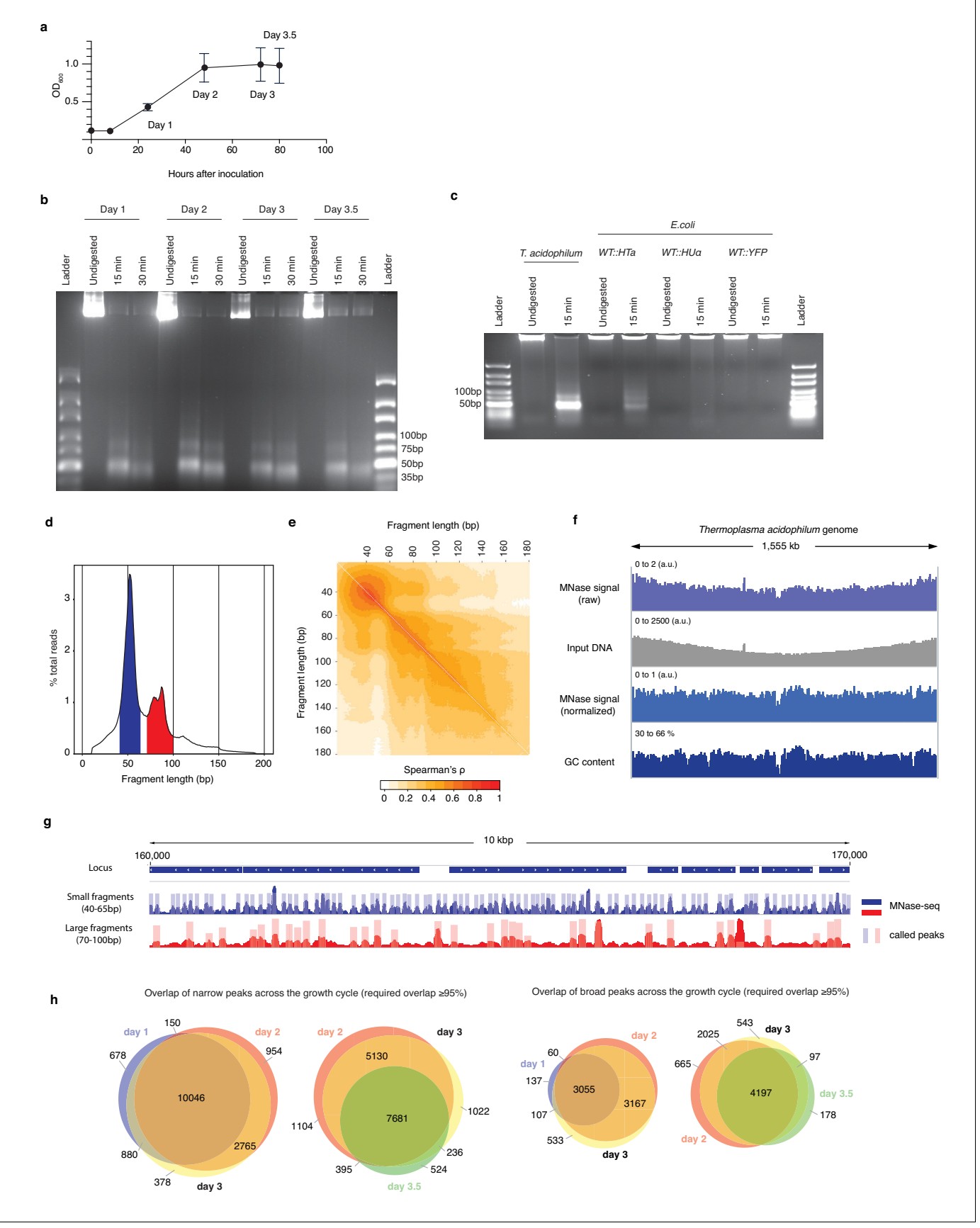

**Figure 3.** HTa-mediated primary chromatin architecture in *T. acidophilum* mapped by MNase-Seq. (a) Growth curve of *T. acidophilum* as determined using optical density ($OD_{600}$). Time points used for downstream experiments are indicated (means and ± SEM across four biological replicates). (b) Agarose gel of MNase digestion products from *T. acidophilum* sampled across the growth cycle. Growth phases are given as days after inoculation, digestion time in minutes. (c) Agarose gel of MNase digestion products from *T. acidophilum* (day 2) along with digestion products of *E. coli* ectopically expressing HTa, HupA or YFP (see Materials and methods). (d) Distribution of the lengths of fragments mapped to the *T. acidophilum* genome (pooled across all four replicates from day 2), highlighting fragment size ranges that correspond to small (blue) and large (red) fragments, as defined in the main text. (e) Correlation matrix comparing genome-wide MNase-Seq coverage signal, computed at base-pair resolution, between reads of defined sizes (pooled replicates, day 2). (f) Genome-wide MNase-Seq signal prior to and after normalization with sonicated DNA input (see Materials and methods), along with GC content profile along the *T. acidophilum* chromosome, computed using a 51 bp moving window. (g) Example of coverage and called peaks across a 10 kb region of the *T. acidophilum* chromosome. (h) Overlap of detected narrow and broad peaks across the growth cycle. Note that different sections/overlaps are only qualitatively but not quantitatively proportional to absolute peak numbers.

The online version of this article includes the following figure supplement(s) for figure 3:

**Figure supplement 1.** Agarose gel (3%) of MNase digestion products from *T. acidophilum* (day 2) along with digestion products of *E. coli* ectopically expressing either HTa, HupA, YFP, HupA (E38K,V42L), HU from *T. composti* or HU from *L. floricola*, from the same plasmid backbone.

**Figure supplement 2.** Distribution of the lengths of fragments mapped to the *T. acidophilum* genome for all replicates across the growth cycle.

**Figure supplement 3.** Heat maps indicating MNase-seq coverage by fragment length relative to the center of broad peaks in *T. acidophilum*, for the same sample (day1, replicate 3), digested for either 15 or 30 min.

**Figure supplement 4.** Multiscale analysis of MNase signal.

from the same plasmid, does not confer significant protection under equivalent digestion conditions (*Figure 3c*). Neither do HU orthologs from *Thermobacillus composti* and *Lactobacillus floricola* (*Figure 3—figure supplement 1*), which have higher sequence identity to HTa (37% and 39% compared to 27% for *E. coli* HupA). HTa is therefore unusual, although perhaps not unique (*Ghosh and Grove, 2004*; *Mukherjee et al., 2008*) among HU family proteins in its capacity to protect DNA from MNase attack. Interestingly, the MNase profile of HTa-expressing *E. coli* suggests protection of additional, longer fragments, not as readily apparent in the *T. acidophilum* digest (*Figure 3c*). We will return to this observation below.

Next, we sequenced the *T. acidophilum* DNA fragments that survived MNase treatment (see Materials and methods). As anticipated, two major fragment size classes are present across all stages of the growth cycle (*Figure 3d*, *Figure 3—figure supplement 2*). For downstream analysis, we define small (large) fragments as 40–65 bp (70–100 bp) in size and note the following: first, the twin peaks centered around ~ 85 bp (*Figure 3d*), separated by approximately one helical turn, were evident across biological replicates (*Figure 3—figure supplement 2*). At present, we do not know whether this reflects the presence of distinct binding species. However, genome-wide occupancy of 70–80 bp and 80–90 bp fragments is highly correlated (Spearman's $\rho = 0.76$, p<2.2e-16, *Figure 3e*) and we therefore consider 70–80 bp and 80–90 bp fragments jointly. Second, modal fragment sizes (~50 bp and ~ 85 bp) are slightly larger than those reported previously (~40 bp and ~ 80 bp) (*Searcy and Stein, 1980*). At least in part, this reflects digestion conditions: fragments obtained after doubling digestion time from 15 to 30 min are shorter and map inside larger footprints (*Figure 3—figure supplement 3*). We chose and persisted with a somewhat milder digest here to avoid over-digestion of small fragments.

We then mapped fragments, irrespective of their size, to the *T. acidophilum* genome (see Materials and methods). Protection across the genome is both ubiquitous and heterogeneous, with relatively even coverage along the origin-terminus axis once increased copy number of early replicating regions is taken into account (see Materials and methods, *Figure 3f–g*). Major drop-offs in coverage correspond to areas of low GC content, as evident from *Figure 3f*, formally shown in *Figure 3—figure supplement 4*, and further explored below. For any given growth phase, genome-wide occupancy is highly correlated across replicates (mean $\rho = 0.89$, p<2.2e-16 for all pairwise comparisons). For each time point, we therefore merged reads across replicates and called peaks independently for small and large fragments (see Materials and methods). Globally, peak locations vary little across the growth cycle (*Figure 3h*). Below, we will focus on data from exponential phase (day 2), where we observe 13,915 narrow and 6887 broad peaks, before discussing variability of HTa-mediated chromatin architecture across the growth cycle in the context of transcriptional changes.

## Analysis of HTa binding footprints suggests histone-like oligomerization behavior

Considering read coverage around called peaks, it is evident that small and large fragments often overlap (*Figure 3g*, *Figure 4c,f*), indicating the presence of different protective entities in the same location across cells. Importantly, broad peaks are typically a combination of small and large fragments and often show asymmetric coverage (caused by smaller fragments) around the summit of the inferred peak (see *Figure 4a* for an example). Given that some archaeal histones (*Maruyama et al., 2013*; *Mattiroli et al., 2017*) and bacterial HU proteins (*Hammel et al., 2016*; *Hołówka et al., 2017*) can form oligomers, we reasoned that asymmetric coverage might contain valuable information about the potential genesis of larger fragments from smaller nucleation sites. To retain this signal, lost when averaging over individual peaks in aggregate plots, we re-oriented the coverage signal as displayed in *Figure 4b*, revealing that smaller fragments are aligned to the edge – rather than the center – of broad-peak footprints (*Figure 4d,g*).

We then applied the same procedure to MNase-Seq data we had independently generated for the histone-containing archaeon *Methanothermus fervidus* (see Materials and methods). Comparing *M. fervidus* to *T. acidophilum,* we find a very similar nested, edge-aligned structure of smaller fragments within broader peaks (*Figure 4e,h*). In *M. fervidus,* this pattern is consistent with oligomerization, in dimer steps, from the minimal histone tetramer (*Maruyama et al., 2013*; *Mattiroli et al., 2017*). Whether the pattern in *T. acidophilum* similarly reflects direct physical contact or is caused by closely adjacent binding of independent HTa complexes remains to be established.

## HTa exhibits histone-like sequence preferences

Next, we asked what factors govern HTa binding in general and presumed nucleation-extension dynamics in particular. As the MNase signal broadly tracks GC content (*Figure 3f*), we first considered nucleotide enrichment patterns associated with peaks. At a coarse level, we find that both broad and narrow peaks exhibit relatively elevated GC content at their center and are flanked by short stretches of GC depletion (*Figure 4i,j*). In line with promiscuous binding, a specific binding motif, as one would observe for most transcription factors, is not evident (*Figure 4—figure supplement 1*). For broader peaks, it is worth noting that, in both *T. acidophilum* and *M. fervidus*, once we have re-oriented the small fragment coverage signal as described above, peak-internal GC content positively tracks small fragment abundance (*Figure 4j,k*), supporting a model where nucleation happens on more GC-rich sites whereas sequence need not be as favorable for subsequent extension events. Note, however, that while sequence might be more important for nucleation than extension, it is by no means irrelevant for the latter: narrow peaks where large fragments are rare tend to be flanked by more AT-rich sequence in both *T. acidophilum* and *M. fervidus*, suggesting that sequence can prevent or at least predispose against oligomer formation (*Figure 4—figure supplements 2–3*).

We then considered dinucleotide frequencies in reads of defined lengths, restricting the analysis to reads that overlap previously defined peaks by at least 90% and discarding duplicate reads that mapped to the same genomic location, so as not to bias results towards highly occupied footprints. Read-internal dinucleotide profiles confirm an overall GC preference but also reveal local enrichment/depletion patterns, notably a short central track of reduced GC enrichment (*Figure 5a*) in *T. acidophilum* as well as histone-encoding archaea, which is symmetric around the HTa/nucleosome dyad. Expectedly, local enrichment patterns get weaker and eventually disappear when considering reads increasingly further away from modal fragment lengths (*Figure 5—figure supplement 1*). Unexpectedly, we also find mononucleotide and RR/YY (R = A or G; Y = T or C) profiles similar to those previously observed for – and attributed to the unique geometry of – eukaryotic histones (*Reynolds et al., 2010*; *Ioshikhes et al., 2011*; *Quintales et al., 2015*). These profiles show strong counter-phasing and asymmetry across the dyad (*Figure 5b,c*) and are particularly prominent in *T. acidophilum* and *Thermococcus kodakarensis*. These observations suggest that symmetric and asymmetric nucleotide enrichments across a dyad axis are not limited to nucleosomes and, furthermore, imply that the HTa-DNA complex is symmetric. All three observations – a general preference for GC-rich sequence, symmetric WW/SS and asymmetric mononucleotide/RR/YY frequencies around the dyad axis – are strongly reminiscent of archaeal as well as eukaryotic histones (*Peckham et al., 2007*; *Kaplan et al., 2009*; *Tillo and Hughes, 2009*; *Ammar et al., 2011*; *Nalabothula et al., 2013*).

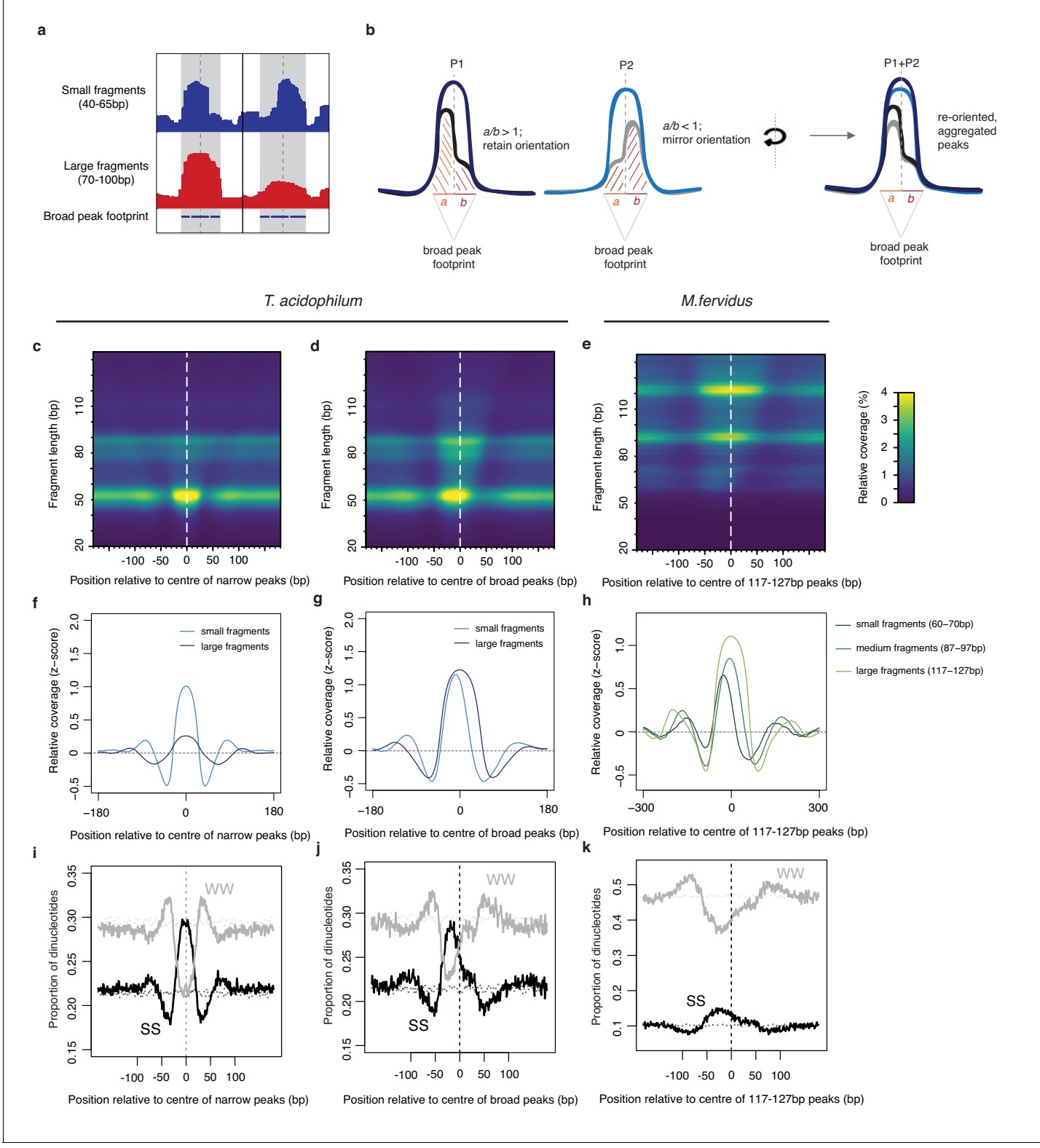

**Figure 4.** Asymmetric coverage signals around peaks in *T. acidophilum* and *M. fervidus* that track underlying nucleotide content. (a) Empirical example and (b) schematic describing our approach to re-orienting coverage signals at broad peaks based on the coverage of small fragments around the dyad axis. (c, d) Heat maps illustrating MNase-seq coverage by fragment length relative to the center of narrow and broad peaks in *T. acidophilum*. Coverage around broad peaks is oriented as explained in (b). (e) Analogous heat map illustrating coverage by fragment length relative to the center of large peaks (corresponding to the binding footprints of octameric histone oligomers) in *M. fervidus*. (f, g, h) Normalized coverage for *T. acidophilum*

*Figure 4 continued on next page*

*Figure 4 continued*

small (40–65 bp) and large (70–100 bp) fragments and *M. fervidus* fragment ranges corresponding to the expected footprint sizes of histone tetramers, hexamers, and octamers. (i, j, k) Proportion of SS (=CC|CG|GC|GG) and WW (=AA|AT|TA|TT) dinucleotides at the same relative positions as (c, d, e). Dotted lines indicate the proportion of SS or WW dinucleotides expected by chance, estimated via random sampling of 25000 regions of equal size in each genome.

The online version of this article includes the following figure supplement(s) for figure 4:

**Figure supplement 1.** Weblogos of bitscores and nucleotide occurrence probabilities at (a) narrow and (b) broad peaks detected during exponential phase in *T. acidophilum*.

**Figure supplement 2.** Normalized MNase-Seq coverage relative to the center of narrow peaks oriented according to the abundance of (a) 87–97 bp fragments in *M. fervidus* and (b) 70–100 bp fragments in *T. acidophilum*.

**Figure supplement 3.** As in *Figure 4—figure supplement 2* but for 87–97 bp peaks scored according to 117–127 bp fragments and oriented according to 60–70 bp fragments.

To define nucleotide preferences more rigorously and enable prediction of relative occupancy from underlying nucleotide features, as previously done for eukaryotic histones (*Tillo and Hughes, 2009*), we trained Lasso models on small and large fragments separately (see Materials and methods). The models confirm a general enrichment for strong (S = G or C) over weak (W = A or T) nucleotides, with mono- and dinucleotide frequencies the strongest individual predictors (*Figure 5e*). For small fragments, predicted and observed occupancy are well correlated ($\rho$(day 2)=0.76; $\rho$(day 3.5)=0.86, observed versus predicted in test set, p<2.2e-16, *Figure 5d*).

## In vivo sequence preferences are also found in vitro and in *E. coli* expressing HTa

Sequence preferences derived from MNase-Seq can be challenging to interpret because micrococcal nuclease activity is characterized by preferential digestion of AT-rich sequences (*Hörz and Altenburger, 1981*; *Chung et al., 2010*; *Allan et al., 2012*). To ensure that the patterns we observe are not simply owing to MNase biases and to better understand the contribution of sequence in determining HTa binding in vivo, we performed a suite of additional experiments.

First, we digested, sequenced, and analyzed naked genomic DNA from *T. acidophilum*. Fragment ends exhibit the expected MNase cutting bias and subtly higher GC than the genomic average (*Figure 5f*). Importantly, however, the read-internal nucleotide enrichment patterns we observe in native chromatin digests (*Figure 5a–c*) are absent from naked DNA (*Figure 5f*), read coverage along the genome is poorly correlated between naked DNA and native chromatin ($\rho$ = 0.07, p<2.2e-16, *Figure 5g*), and a Lasso model trained on naked DNA fails to predict occupancy in vivo (*Figure 5—figure supplement 2*, $\rho$ = −0.04, p<2.2e-16).

Second, we surveyed digest fragments obtained from HTa-expressing *E. coli* and compared them to fragments from an *E. coli* genomic DNA digest. Reads obtained from the HTa-expressing strain exhibit internal nucleotide enrichments matching those in *T. acidophilum*. Reads from the genomic DNA do not (*Figure 5f*). Neither do reads from an *E. coli* native chromatin digest (*Rojec et al., 2019*). In addition, Lasso models trained on fragments f HTa-expressing *E. coli* predict occupancy in *T. acidophilum* (*Figure 5—figure supplement 2*, rho = 0.54, p<2.2e-16). These findings further argue against MNase bias as a major confounder and bolster the conclusion that the signal observed in *T. acidophilum* chromatin is chiefly driven by HTa.

Third, we purified untagged HTa recombinantly expressed in *E. coli* (*Figure 6—figure supplement 1*) to reconstitute *T. acidophilum* chromatin at a physiological HTa:DNA mass ratio of 0.4 (see Materials and methods). We then treated the resulting nucleoprotein assemblies with MNase. As in prior in vitro work (*Searcy and Stein, 1980*), this revealed protected fragments of defined lengths similar to what is seen in vivo (*Figure 6—figure supplement 1*). Coverage of small fragments in vitro and in vivo is well correlated (*Figure 6a*, $\rho$ = 0.61, p<2.2e-16) and the Lasso model trained on the in vitro data predicts occupancy in vivo (*Figure 6b*, $\rho$ = 0.68, p<2.2e-16). This is consistent with a major role for sequence in driving positioning of HTa in vivo, with 42% of the variance in occupancy in vivo explained by sequence preferences in vitro.

Finally, we carried out electrophoretic mobility shift assays (EMSAs), which do not involve MNase treatment and are therefore free from digestion biases. We selected five 100 bp regions across the *T. acidophilum* genome with very low to very high (5th, 25th, 50th, 75th, 95th percentile) occupancy in

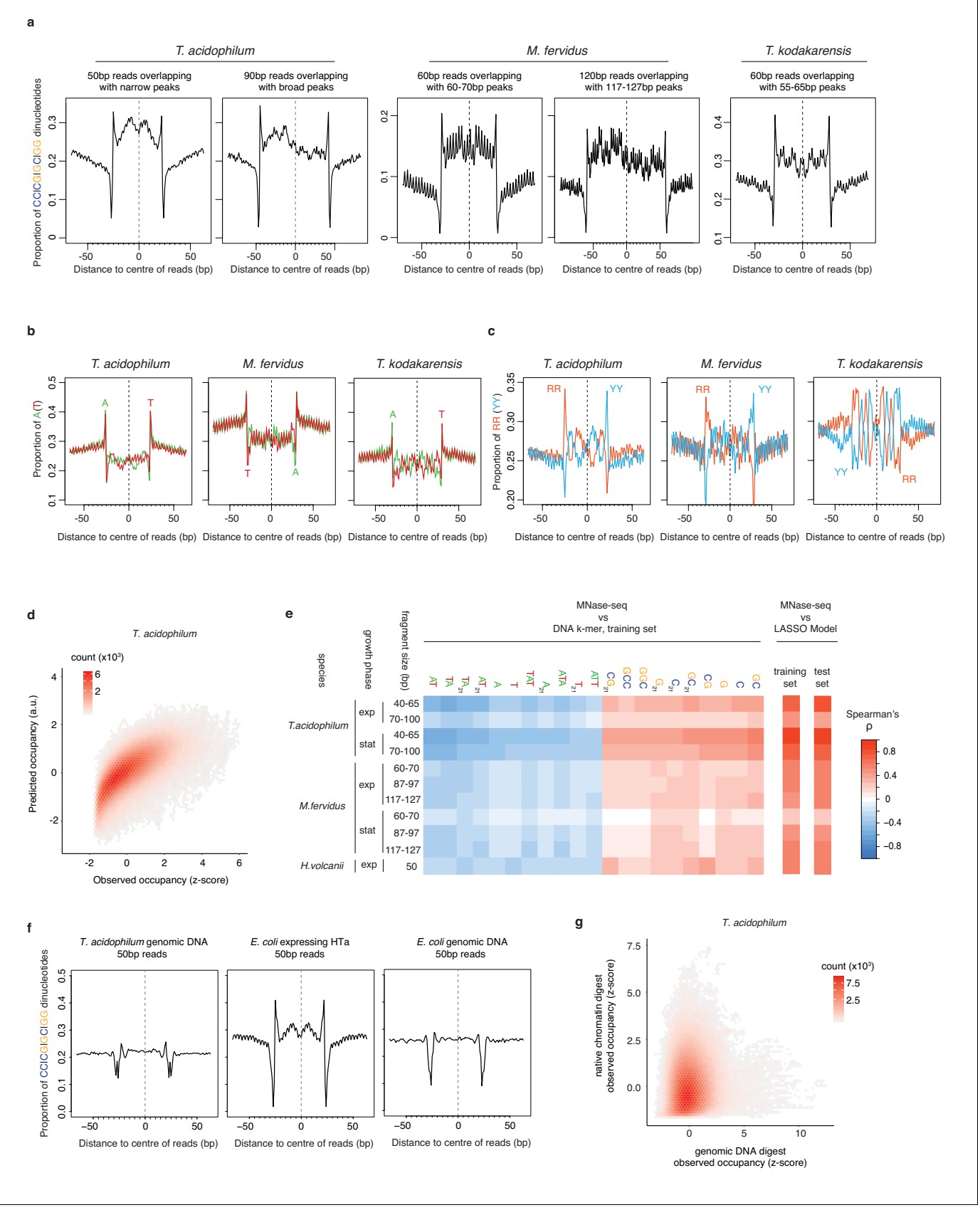

**Figure 5.** Comparison and predictive power of nucleotide enrichment patterns associated with HTa and archaeal histones. (a) Proportion of SS (=CC| CG|GC|GG) dinucleotides, (b) A|T mononucleotides, and (c) RR (=purine/purine)|YY (=pyrimidine/pyrimidine) dinucleotides relative to the centers of reads of defined length in different archaeal species (see Materials and methods for read filtering). (d) Density plot comparing observed (day 2, replicate 2) and predicted MNase-Seq coverage across the part of the *T. acidophilum* chromosome not used for training. (e) Correlation between MNase-seq coverage and individual DNA k-mers with particularly high positive or negative correlation coefficients, as observed in the training data. Overall correlations between measured MNase-Seq coverage and coverage predicted by the LASSO model, for both trained and untrained data, are shown on the right-hand side. (f) Proportion of SS dinucleotides relative to the centers of 50 bp reads from digests of *T. acidophilum* genomic DNA, *E. coli* expressing HTa, and *E. coli* genomic DNA. (g) Genome-wide correlation of normalized occupancy between *T. acidophilum* genomic DNA and native chromatin digests.

The online version of this article includes the following figure supplement(s) for figure 5:

**Figure supplement 1.** Proportion of SS (=CC|CG|GC|GG) dinucleotides relative to the centers of reads of defined length (41–53 bp) in *T. acidophilum*.
**Figure supplement 2.** Predicting in vivo HTa occupancy.

vivo and similarly low/high occupancy in vitro (*Figure 6—figure supplement 2*). These 'backbone' oligos were then further diversified by introducing eight random dinucleotides at defined positions along the backbone to generate a large library of double-stranded DNA oligos with variable nucleotide content (see Materials and methods, *Figure 6—figure supplement 2*). Addition of HTa retards the migration of a subset of dsDNA oligos in a concentration-dependent manner (*Figure 6c,d*). At high HTa:DNA mass ratios, the majority of oligos are subject to retardation, as expected from proteins that, like histones, have compositional preferences but are capable of binding most sequences. To understand HTa binding preferences, we need to investigate differences at low HTa:DNA stoichiometry. To identify sequences preferentially bound by HTa when HTa is limiting, we therefore excised and sequenced individual bands from two replicate experiments carried out at an HTa:DNA mass ratio of 0.2 (see Materials and methods, *Figure 6d*). To determine whether the different oligo backbones act as anticipated even outside their native genomic context, we considered the proportion of reads derived from a given backbone that were found in the slowest-migrating band ('HTa-bound slow' in *Figure 6d*). The expectation here is that backbones with higher occupancy in vivo and in vitro should be bound more readily or with greater affinity by HTa and therefore be relatively enriched in the slowest-migrating band (quantified as the proportion of reads of a given type, $P_{slow}$, found in the HTa-bound slow band, see Materials and methods). The data confirm this expectation (*Figure 6—figure supplement 2*), with the exception of the most AT-rich oligo, which exhibits unexpectedly high $P_{slow}$ (*Figure 6—figure supplement 2*), a phenomenon previously also observed for eukaryotic histones (*Tillo and Hughes, 2009*). We exclude sequences derived from this backbone, whose low occupancy in vivo and in vitro might be contingent on wider sequence context, from further analysis. For the remaining oligos, we then ask if $P_{slow}$ varies as a function of nucleotide content, as predicted by our MNase-Seq data. We find this to be the case. $P_{slow}$ is positively correlated with GC content (*Figure 6e*, *Figure 6—figure supplement 3*) and, even more strikingly, with the number of GpC dinucleotides in the oligo (*Figure 6f*, *Figure 6—figure supplement 4*). The latter is noteworthy because GpC dinucleotide content is the strongest individual predictor in Lasso models for in vivo chromatin occupancy (*Figure 5e*). This orthogonal experiment corroborates that HTa has an unusual, histone-like preference for GC-rich DNA.

## A hidden diversity of large fragments in exponential phase

As demonstrated above, the Lasso model trained on in vivo chromatin predicts small fragment occupancy across the growth cycle well ($\rho > 0.7$, *Figure 5e*). Curiously, it performs much worse when trying to predict large fragment coverage in exponential phase (*Figure 5e*, day 2, $\rho = 0.43$, p<2.2e-16). Why would this be? We reasoned that larger fragments might come from a mixture of protective binding events. For example, in addition to nucleation-extension events, larger fragments might results from a different mode of HTa binding that is independent of prior sequence-driven nucleation ('independent fragments'). Alternatively, larger fragments may reflect the binding of an altogether different protein or protein complex that happens to protect a similar-sized piece of DNA. To explore this hypothesis, we divided broad peaks into deciles based on the relative coverage of small fragments at these peaks. Doing so, we find that broad peaks where small fragments are rarest show strongly divergent sequence composition (*Figure 7a*). These divergent peaks are common in exponential phase, but disappear almost entirely in stationary phase, which is dominated by broad

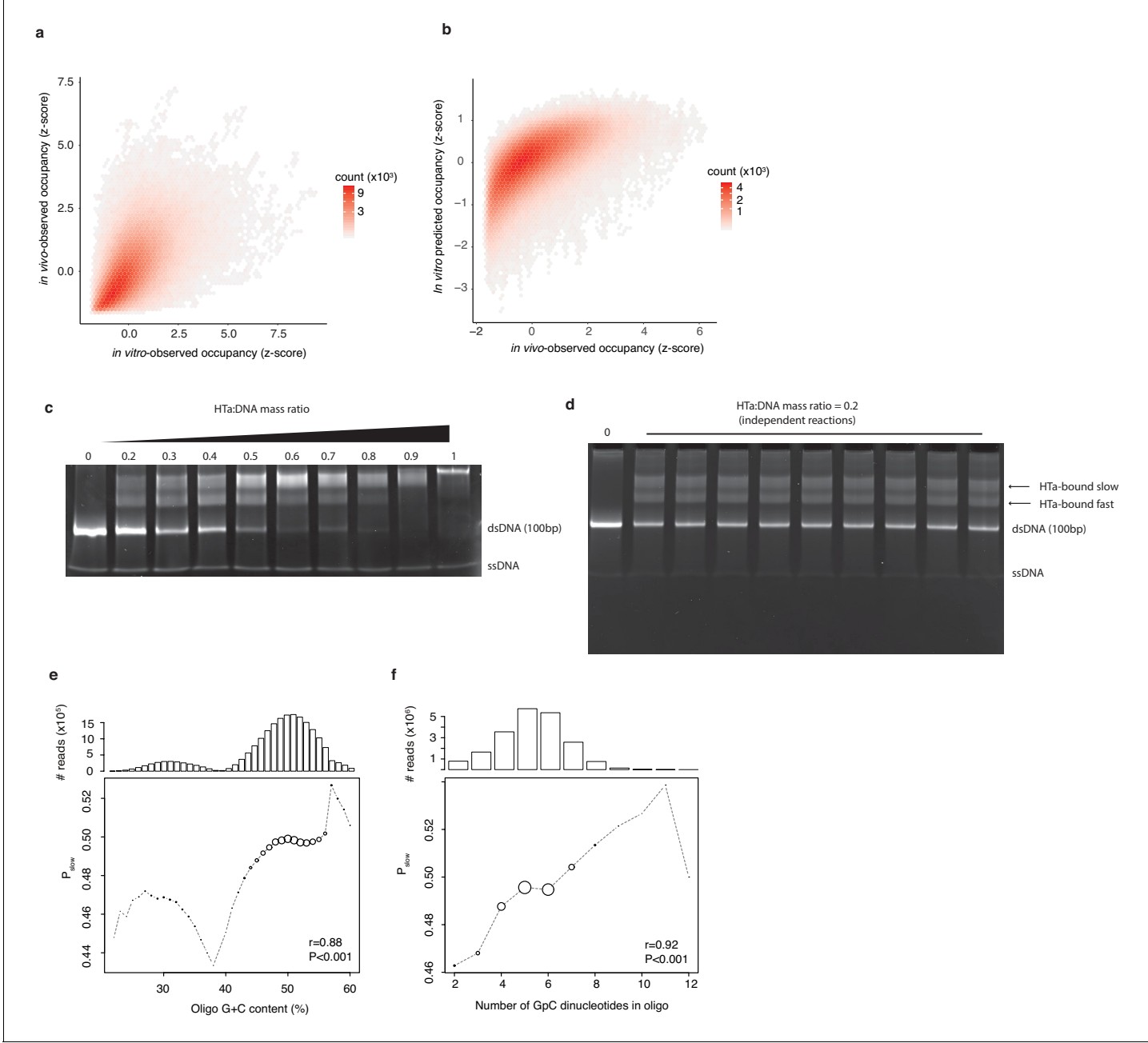

**Figure 6.** In vitro experiments to assess HTa binding preferences. (**a**) Occupancy of small fragments across the *T. acidophilum* genome in vivo (day 2) correlates with occupancy following in vitro reconstitution and with (**b**) occupancy predicted by a Lasso model trained on the in vitro data. (**c**) EMSAs on libraries of sequence-variable dsDNA oligomers (see main text) in the presence of increasing amounts of HTa. (**d**) Independent reactions at a HTa:DNA ratio of 0.2 yield highly reproducible band shift patterns. (**e**) $P_{slow}$ varies as a function of oligo G+C content and (**f**) GpC dinucleotide content. Point sizes are scaled according to the relative abundance of reads of a given G+C (GpC) content across the sequenced bands. The absolute number of reads analyzed is given in the panel above. Correlation coefficients (r) are from Pearson correlations between G+C (GpC) content and $P_{slow}$ weighted by the number of reads at each G+C (GpC) content.

The online version of this article includes the following figure supplement(s) for figure 6:

**Figure supplement 1.** In vitro reconstitution of HTa:DNA nucleoprotein complexes.

**Figure supplement 2.** EMSA backbone sequences.

**Figure supplement 3.** The relationship between GC content of an oligo and $P_{slow}$.

**Figure supplement 4.** The relationship between GpC dinucleotide content of an oligo and $P_{slow}$.

peaks that conform to the nucleation-extension model instead (*Figure 7b*, *Figure 7—figure supplement 1*). Broad peaks with few small fragments are particularly enriched in intergenic sequence (*Figure 8a*, hypergeometric test p<2.2e-16), and it is around transcriptional start sites (TSSs, *Figure 8b*; approximated from RNA-seq data, see Materials and methods) and end sites (TESs, *Figure 8—figure supplement 1*), where their disappearance in stationary phase is most striking (*Figure 8b*). We find no equivalent to these independent large fragments in histone-encoding archaea (*Figure 8c–e*, *Figure 8—figure supplement 1*) suggesting that they are specific to *T. acidophilum*. Neither in vitro reconstitution nor HTa expression in *E. coli* yielded similar independent fragments. It is therefore parsimonious to conclude that these fragments likely reflect the binding of different proteins or protein complexes in *T. acidophilum* whose identity remains to be established.

Importantly, even though their positioning suggests that they might be involved in gene regulation, the disappearance of independent fragments in stationary phase is not obviously coupled to local changes in transcription: the abundance of independent fragments drops in similar fashion for genes that are up- or down-regulated or remain the same in stationary compared to exponential phase (*Figure 8f*). At the same time, the relative abundance of smaller fragments around transcriptional start sites appears entirely insensitive to changes in transcription (*Figure 8f*), and thus provides no support for a role of HTa in the dynamic regulation of transcription.

## Discussion

The evidence above suggests that HTa is a protein of bacterial origin that converged onto supramolecular properties reminiscent of archaeal histones. Whereas well-characterized HU homologs exhibit elevated occupancy in AT-rich regions, prefer an AT-rich motif at the center of the binding footprint, or bind sequence non-selectively (*Swinger and Rice, 2007*; *Grove, 2011*; *Prieto et al., 2012*), HTa

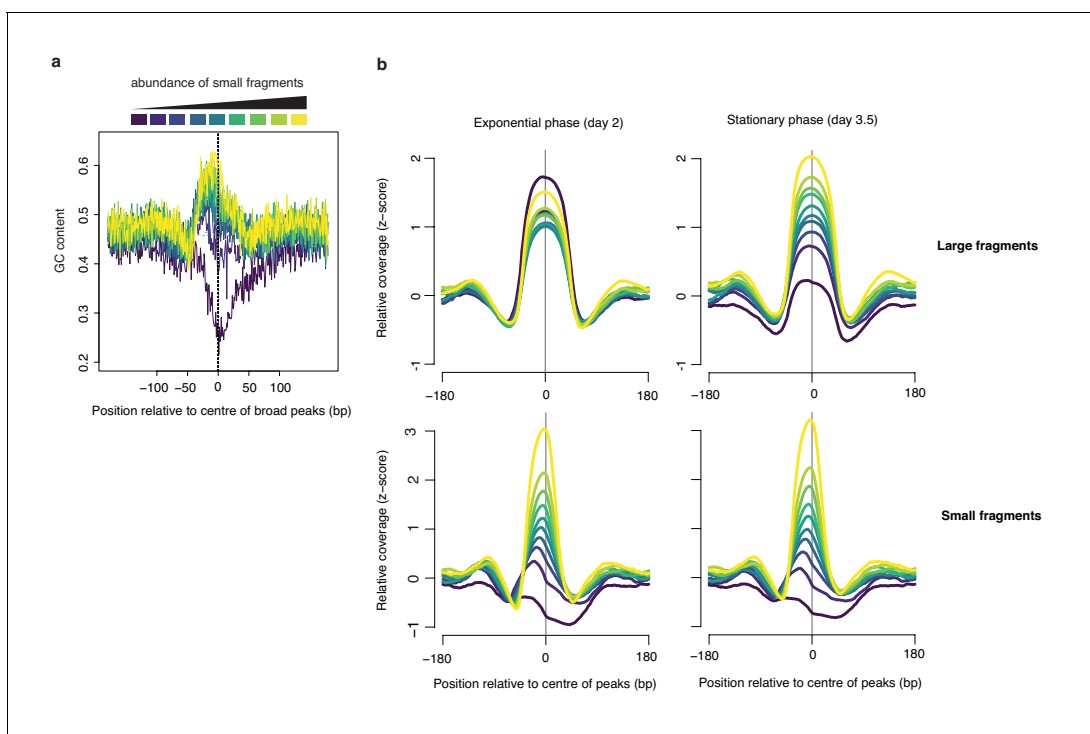

**Figure 7.** Broad peaks are associated with heterogeneous GC content in exponential but not stationary phase. (**a**) Average GC content at broad peaks (day 2), separated into deciles based on the relative abundance of small fragments and (**b**) the corresponding relative coverage for large and small fragments during exponential and stationary phase. For all graphs, decile decomposition is based on small fragment occupancy during exponential phase (day 2).

The online version of this article includes the following figure supplement(s) for figure 7:

**Figure supplement 1.** Small fragment abundance at narrow peaks.

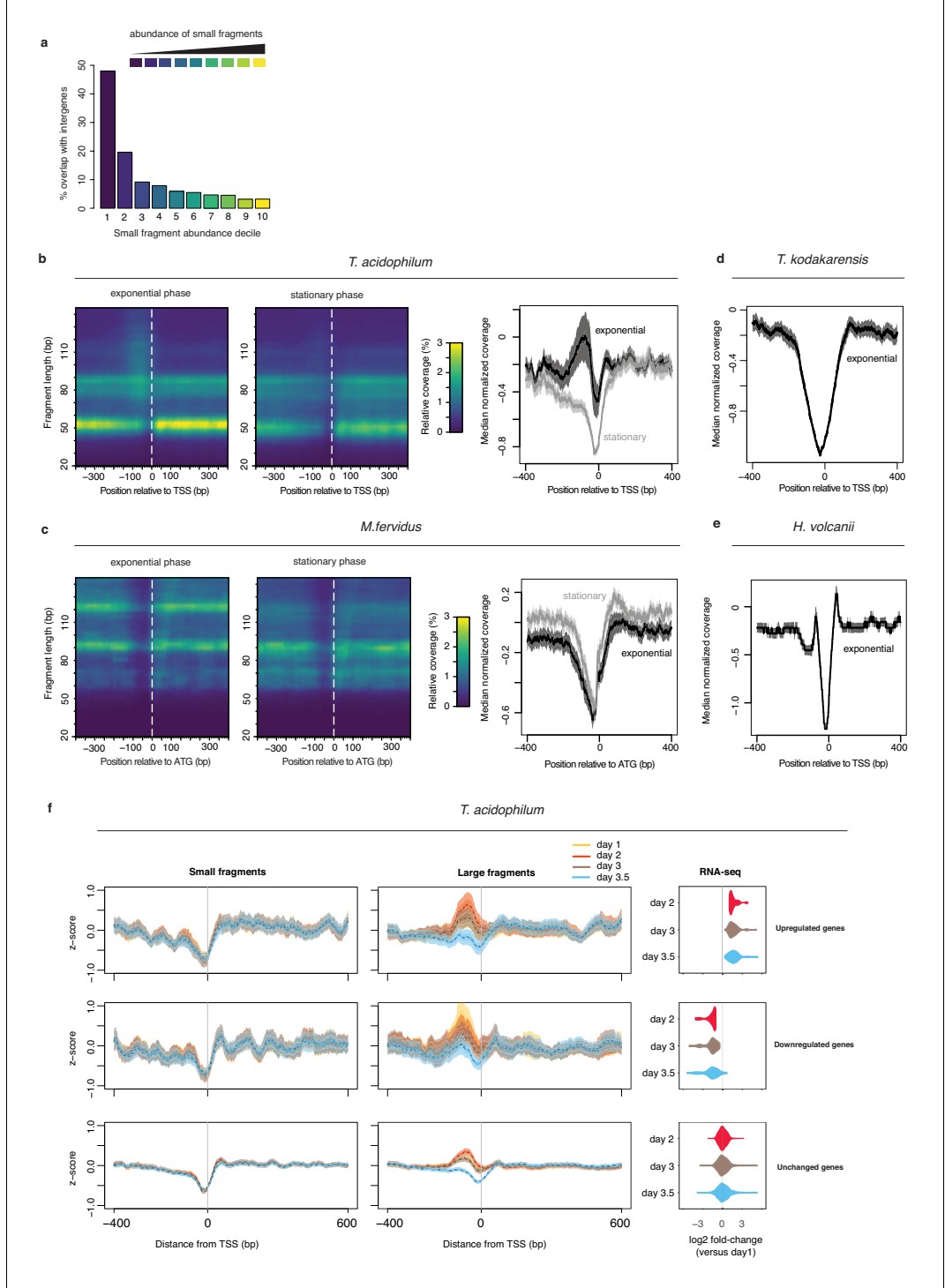

**Figure 8.** MNase-Seq coverage around transcriptional start sites in *T. acidophilum* and histone-encoding archaea in the context of dynamic transcription. (a) Broad peaks associated with low abundance of small fragments are enriched in intergenic regions. (b) Left and central panel: Heat maps indicating MNase-seq coverage by fragment length relative to transcriptional start sites in exponential (day 2) and stationary phase (day 3.5). Right panel: median normalized MNase-seq coverage (considering all fragment sizes) as a function of distance from the transcriptional start site (TSS). (c) as in (b) but for *M. fervidus* and using the coding start (ATG) rather than the TSS as a reference point. To ensure that the coding start constitutes a reasonable proxy for the TSS, only genes with a divergently oriented neighboring gene are considered, thus eliminating genes internal to operons. (d, e) median of normalized MNase-seq coverage (considering all fragment sizes) as a function of distance from the TSS in *T. kodakarensis* and *Haloferax volcanii*. (f) Changes in normalized MNase-seq coverage for small and large fragments around transcriptional start sites in *T. acidophilum* as a function of growth phase and whether genes are upregulated, downregulated or remain unchanged relative to mRNA abundance on day 1. Genes are grouped

*Figure 8 continued on next page*

*Figure 8 continued*

according to differential expression (or lack thereof) on day two compared to day 1. Genes with a log2-fold change > 1 were considered significantly upregulated, those with a log2-fold change <-1 significantly down-regulated (FDR < 0.01). The rightmost panels indicate that a majority of genes up-/downregulated on day 2, remain up-/downregulated on days 3 and 3.5.

The online version of this article includes the following figure supplement(s) for figure 8:

**Figure supplement 1.** HTa and histone occupancy around transcription end sites.

shows a more histone-like preference for GC and even exhibits asymmetric mononucleotide/RR/YY profiles across the dyad. The in vitro experiments we present above make a cogent argument for sequence as a principal determinant of HTa binding in vivo. Sequence-driven positioning around functional genomic landmarks, footprint extension dynamics, and relative positional stability throughout the growth cycle are also reminiscent of archaeal histones, as is the ability of HTa to protect DNA from MNase digestion. Based on these observations, we propose that HTa can be regarded as an archaeal histone analog.

This analogy is, of course, preliminary. Structural work will be required to characterize how HTa interacts with DNA and to determine whether large HTa-protected fragments reflect closely spaced binding events or, in further analogy to archaeal histones, the presence of contiguous HTa polymers. The observation that even larger protected fragments can be formed when HTa is expressed in *E. coli* (*Figure 3—figure supplement 1*) or reconstituted in vitro (*Figure 6—figure supplement 1*) is interesting in this regard and deserves further exploration. Additional work is also required to elucidate the interaction partners and physiological functions of HTa, including its involvement in transcription. Although we find no obvious global link between changes in HTa occupancy and transcriptional output, this does not preclude local effects or dynamics at a time-scale where information on nascent transcription – rather than steady-state RNA levels – would be required to implicate HTa. Neither do our results rule out an important but constitutive role in transcription, for example in binding to and constraining negative supercoils similar to what has been observed for other HU homologs (*Berger et al., 2010*; *Grove, 2011*) and archaeal chromatin proteins such as MC1, Sul7, and Cren7 (*Zhang et al., 2012*). Of course, the main function(s) of HTa might simply not lie in transcription. Searcy and co-workers showed that HTa facilitates re-annealing following DNA denaturation (*Stein and Searcy, 1978*), suggesting that HTa might assist DNA renaturation after thermal stress. Our finding of heterogeneous but predictable HTa occupancy across the genome implies that re-annealing would proceed differentially in genic and intergenic regions, leaving the latter unbound and free to engage in binding of protein complexes for transcription. However, the physiological importance of HTa-mediated reannealing remains to be established and is put into perspective by the observation that *E. coli* HU, when added to naked DNA, also significantly raises its melting temperature (*Drlica and Rouviere-Yaniv, 1987*). Another possibility is that HTa is involved in higher order genome structure (*Bohrmann et al., 1990*) or, like many HU family members (*Grove, 2011*), in DNA repair. To dissect these and other hypothetical biological functions in vivo in the future and address whether archaeal histones can functionally substitute for HTa, the development of genetic tools for *T. acidophilum* is highly desirable.

Finally, from an evolutionary perspective, it is worth highlighting dinoflagellates as a second case where histones have lost their pre-eminent role in genome compaction and organization to other small basic proteins. Even though histones remain encoded in dinoflagellate genomes (*Marinov and Lynch, 2016*), and might continue to play important roles in transcription or other processes, they are no longer the main protein constituent of chromatin. Instead, proteins with homologs in phycodnaviruses have become the principal mediators of compaction (*Gornik et al., 2012*). In a subset of species, these dinoflagellate/viral nucleoproteins (DVNPs) act alongside HU-like proteins that were likely acquired from bacteria (*Figure 2a*) (*Wong et al., 2003*; *Janouškovec et al., 2017*) and might be involved in the genesis and maintenance of DNA loop structures (*Chan and Wong, 2007*). If not exactly a precedent, the case of dinoflagellates nonetheless serves to illustrate that HU proteins are versatile, evolvable and have been independently co-opted into important chromatin architectural roles following horizontal transfer. It also highlights that histones or a subset of their functions can be replaced, under the right circumstances, by alternative DNA-binding proteins with fundamentally different folds. At the

same time, our results demonstrate that such replacements, even though they appear radical, need not necessarily go hand in hand with fundamental changes to the architectural layout of chromatin.

## Materials and methods

### Thermoplasma acidophilum *culture*

*T. acidophilum* strain 122-1B2 was obtained from DSMZ (https://www.dsmz.de/) and cultured using the medium described in *Searcy and Stein (1980)*, supplemented to a final concentration of 2 g/L yeast extract (BD Biosciences). The medium was boiled for five minutes and allowed to cool to 58°C before inoculation. Cultures were incubated at 59°C with shaking (90 rpm) in an INFORS Thermotron incubator. Throughout, culture to flask volume ratio was maintained at a maximum of one fifth. Fresh cultures were inoculated with a 10% v/v inoculum from a 4-day-old culture. Samples from 24-, 48-, 72-, and 80 hr cultures were used for RNA extraction and MNase experiments. Sample aliquots were first equilibrated to pH4 using $NH_4OH$ to avoid depurination (*Robb et al., 1995*) and then pelleted at low speed. *T. acidophilum* cells tend to filament under stress or in late stationary phase. Cells were therefore imaged with a standard light microscope at 100x magnification to monitor the health of the culture.

### MNase digestion – Thermoplasma acidophilum

*T. acidophilum* cells lyse at pH > 6 (*Robb et al., 1995*). Pellets were therefore directly re-suspended in ice-cold MNase digestion buffer (10 mM Tris, 5 mM $Ca^{2+}$, pH8) and homogenized via 20 passages through a Dounce homogenizer, on ice. Unfixed crude lysates were digested in the presence of 4 U/mL of MNase (Thermo Fisher) at 37°C. Digestion was stopped by addition of EDTA to a final concentration of 20 mM. Samples were incubated for an additional 30 min at 37°C in the presence of RNAse A to a final concentration of 0.5 mg/mL and then overnight at 65°C in the presence of SDS (1%) and proteinase K (125 µg/mL). Subsequently, DNA was extracted by phenol chloroform extraction and precipitated with ethanol. MNase digestion of naked *T. acidophilum* genomic DNA was done using a ratio of 0.006U MNase per 10 µg of DNA, at 37°C for 30 min.

### Preparation of undigested DNA samples

Undigested lysate was incubated as above but without addition of enzyme. Undigested DNA was then sonicated using a Covaris S220 sonicator with the following settings: peak power: 175, duty: 10, cycle/burst: 200 for 430 s (target size: 150 bp).

### MNase digestion and sequencing – Methanothermus fervidus

Flash-frozen pellets of *M. fervidus* harvested in late exponential and stationary phase were purchased from the Archaeenzentrum in Regensburg (via Harald Huber). Digestion and sequencing conditions are described in *Rojec et al. (2019)*. MNase-Seq data associated with this study have been deposited at the NCBI Gene Expression Omnibus (GEO GSE127678).

### MNase digestion and sequencing – Escherichia coli

Pellets were obtained as follows: a fresh culture was inoculated at OD = 0.05 and grown in LB for 2 hr, at which point rhamnose was added to a final concentration of 5 mM and the culture grown for an additional 4 hr. Ten mL of culture was pelleted and snap frozen to be used as starting material for the MNase experiments. For MNase digestion of *E. coli* samples, lysates were obtained by cryo-grinding with a pestle and mortar. The protocol used was similar to the one used for *T. acidophilum* except that intact cells were removed by centrifugation after the RNase digestion step. MNase digestion of naked *E.coli* genomic DNA was done using a ratio of 0.006U MNase per 10 µg of DNA, at 37°C for 30 min.

### RNA extraction

Pellets were re-suspended in 500 µL RNAlater (Ambion), incubated for one hour at room temperature, pelleted again and snap-frozen. Total RNA was extracted using an RNeasy kit (Qiagen) according to manufacturer's instruction, including DNAse I treatment.

## Protein expression in E. coli

Protein sequences for HTa and different HU homologs were obtained from UniProt (P02345, P0ACF0, L0EKC1, A0A0R2DT96). The corresponding coding sequences were codon-optimized for expression in *E. coli*, synthesized and inserted into the pD864-SR backbone for rhamnose-inducible expression (ATUM). A YFP-expressing control plasmid was obtained directly from ATUM and transformed into an *E. coli* C600 (a kind gift from Jacques Oberto).

## Protein purification

HTa was purified from the same *E.coli* strain used for MNase experiments. The purification process was adapted from *Orfaniotou et al. (2009)*, who describe the purification of the HTa ortholog HUtvo from *Thermoplasma volcanium*. Protein purity was confirmed both by denaturing protein gel electrophoresis and by mass-spectrometry. Protein concentration was measured using Bradford and Lowry assays and confirmed by gel electrophoresis using BSA as standard. Purified protein was stored in a buffer containing 50% glycerol, 1 M NaCl, 50 mM Tris at $-80°C$.

## In vitro reconstitution

In vitro reconstitutions were carried out using genomic DNA from *T. acidophilum*, purified by phenol chloroform extraction, and HTa, purified as described above using the protocol of *Searcy and Stein (1980)*. Briefly, 50 μg of DNA were mixed with 20 μg of HTa and volume was reduced to 50 μL using an Eppendorf Concentrator vacuum dryer. The obtained solution was adjusted to 6M Urea 2.5M NaCl 0.25 mM EDTA. Urea and salt were removed by dialysis using Slide-A-Lyzer Mini Devices, 3.5K MWCO (0.5 mL) at 4°C. The obtained reconstituted chromatin was equilibrated to 5 mM Ca2+ before MNase digestion. MNase digestion was done on 5 μg of DNA (measured by nanodrop on the reconstituted chromatin) as described above, but omitting the RNAse step.

## EMSA-seq

Randomized DNA oligonucleotides based on the backbones in *Figure 6—figure supplement 1* were purchased from Sigma and annealed using a thermocycler following Sigma's procedure (https://www.sigmaaldrich.com/technical-documents/protocols/biology/annealing-oligos.html). For EMSA, fresh HTa aliquots were mixed with a pool of randomized oligos (500 ng per reaction) in a HTa:dsDNA mass ratio of 0.2 in a buffer containing a final concentration of 1% glycerol, 20 mM NaCl, 30 mM Tris-Hcl, and 0.7 mM EDTA in a final volume of 24 μL. 4 μL of 6x loading buffer (without SDS, NEB #B7025S) were added just before gel electrophoresis. For each replicate 10 reaction were run on the same gel (12% polyacrylamide, see *Figure 6*) at 90V for 70 min at 4°C, in a pre-chilled 10 mM Tris pH8 buffer. DNA was stained with EtBr and the gel washed twice before imaging. After imaging, the two visibly shifted bands were separately extracted from the gel. For each replicates, material from across the 10 reactions was pooled and DNA extracted following the crush and soak method. DNA concentration was measured by Qbit and a minimum of 10 ng of DNA was used for library preparation and sequencing.

## Bacterial strains and growth conditions

All strains were grown in LB medium at 37°C with shaking. For MNase assays, strains were pre-cultured from selection plates to 5 mL LB cultures in 50 mL falcon tubes overnight with ampicillin at a final concentration of 0.1 mg/mL and re-suspended to OD600 = 0.1. 1M rhamnose was added two hours after inoculation to a final concentration of 5 mM and grown for 4 hr prior to harvesting. Induction was confirmed by protein gel electrophoresis and by monitoring YFP expression by eye.

## Sequencing and data availability

For MNase digestion experiments paired-end reads were prepared using the NEBNext Ultra II DNA Library Prep Kit. For RNA sequencing, ribosomal RNAs were depleted using Illumina's Ribo-Zero rRNA Removal Kit (Bacteria) and RNA libraries prepared using the TruSeq Stranded Total RNA LT Kit. Both RNA and DNA libraries were then sequenced on a HiSeq2500 machine. Raw read and processed data have been deposited in GEO (GSE127728).

## MNase-seq analysis

Paired-end 100 bp reads were first trimmed using BBDuk v37.64 (parameters: ktrim = r, k = 21, hdist = 1, edist = 0, mink = 11, qtrim = rl, trimq = 30, minlength = 10, ordered = t, qin = 33) and merged using BBmerge v37.64 (parameters: mininsert = 10, mininsert0 = 10). The merged reads were then mapped to the *T. acidophilum* DSM1728 genome (GCA_000195915.1) using Bowtie2 (-U). Coverage tracks were computed using bedtools bamCoverage (RPGC normalization, effective genome size: 1564906 bp), measuring coverage for reads of sizes 40–65 bp and 70–100 bp separately where appropriate. To compute the correlation matrix of reads of different sizes (*Figure 3e*), coverage for reads of defined lengths was computed using bedtools bamCoverage without RPGC normalization.

## Normalization for replication associated bias

Reads from the sonicated DNA control samples were mapped in paired-end mode using Bowtie2. Coverage was computed independently of read size and smoothed over 10 kbp to avoid introducing noise from the input into the MNase signal. Genome-wide coverage of MNase-digested samples was then divided by its corresponding undigested DNA sample to remove bias in coverage associated with replication. Lastly, coverage was converted into a Z-score using the *scale* function in R. When reads from inputs were not available (*E. coli* experiments and *T. acidophilum* chromatin reconstitution), replication associated bias was removed by self-normalizing coverage tracks with a 250 kb smoothed track computed on the same signal.

## RNA-seq analysis

Single-end reads were mapped to the *T. acidophilum* DSM1728 genome using Geneious 11.1.2. Transcripts were reconstructed and quantified using Rockhopper (*McClure et al., 2013*), using a stringent threshold of 0.9 for 5' and 3'UTR detection. Differential expression was assessed using DESeq2 as part of the Rockhopper pipeline.

## LASSO modeling

Abundances of different nucleotide k-mers k={1,2,3,4} were computed over genomic windows of 21 bp and 51 bp using the R package seqtools (v1.16). We included the 21 bp window to enable independent capture of read-internal nucleotide patterns that did not overlap with MNase cutting sites. For all genomes analyzed, the top 80 k-mers (across both 21 bp and 51 bp windows) with the highest absolute correlation values with MNase coverage over the first third of the genome were chosen as model parameters. A general linear model with LASSO optimization was trained on the first third of each genome, with 10-fold internal cross validation, using the LASSO function in Matlab. Predicted coverage tracks were then calculated in R based on k-mer weights from the training set.

## Peak detection and asymmetry scoring

Peaks were detected using the NucleR R package and scored using a modified scoring function, where peak height was measured as the coverage value at each peak center relative to the empirical distribution of the data. Parameters used for the initial Fourier filtering step are listed in *Supplementary file 2*. A threshold value corresponding to a score of 2.5 (see NucleR manual) was used for all data. To score asymmetric coverage inside broad peaks, we computed the average coverage signal of smaller DNA fragments either side (*a* and *b* in *Figure 4b*) of the peak dyad and then considered the ratio $a/b$. Coverage signal at peaks where $a/b < 1$ were flipped around the dyad axis for downstream signal processing.

## Nucleotide frequencies and data visualization

Nucleotide frequencies were computed using the R package Seqpattern (v1.14). Only reads with an average quality score of 30 that overlapped called peaks by $\geq$ 90% were selected for this analysis. 2D occupancy plots were generated from non-normalized MNase-Seq reads using plot2DO (https://github.com/rchereji/plot2DO) (*Chereji et al., 2018*), modified to enable processing of single-end reads. Multiscale analysis of MNase coverage was carried out using MultiScale Representation of Genomic Signals (https://github.com/tknijnen/msr/), considering genomic segments with significant (p<1e-10) enrichment at scale 30 (see *Knijnenburg et al. (2014)* for a detailed description of the statistical approach).

## MNase and additional data from other organisms

MNase data from other organisms was obtained from the Sequence Read Archive (SRR495445 for *T. kodakarensis*; SRR574592 for *H. volcanii*). *H. volcanii* reads were trimmed using BBDUK (ktrim = r, k = 21, hdist = 1, edist = 0, mink = 11, qtrim = rl, trimq = 20, minlength = 10, qin = 33). Reads were mapped to their respective genomes using Bowtie2 (Setting for SRR495445 reads: −3 75 -X 5000 k 1 -x, as in the original publication; default settings for SRR574592 reads). To reduce PCR duplicate bias, per base coverage values of MNase data from *H.volcanii* were thresholded at the last percentile. TSS positions for *T. kodakarensis* and *H. volcanii* were obtained from *Jäger et al. (2014)* and *Babski et al. (2016)*, respectively.

## Homology modeling

Secondary structures for HupA (P0ACF0) and HmfA (P48781), as displayed schematically in *Figure 1a*, were taken from UniProt. The secondary structure of HTa was predicted by homology modeling in SWISS-MODEL (*Waterhouse et al., 2018*) using HupA (PDB:1mul) as a template. To predict and visualize the quaternary structure of the HTa homodimer, we used the HTa sequence to build a homology model based on the X-ray crystal structure of (HupA)$_2$ using PDB:1p51 as a template. The homology model, again built using SWISS-MODEL, has a general mean quality estimate of 0.71. Both (HTa)$_2$ and (HupA)$_2$ structures were refined with steepest descent and conjugate gradient energy minimization using the AMBER ff14SB protein force field potentials (*Maier et al., 2015*) and a force constraint of 2 kcal/mol placed on the Cα peptide backbone atoms. To calculate the solvent accessible surface area and charge density, we used the Adaptive Poisson-Boltzmann Solver (APBS) (*Jurrus et al., 2018*). The charge density was mapped onto the solvent accessible surface area using the VMD visualizsation package (*Humphrey et al., 1996*).

## Phylogenetic analysis

Amino acid sequences containing the HU-IHF domain (cl00257) where identified in bacteria, archaea, eukaryotes, and viruses using the Conserved Domain Architecture Retrieval Tool (*Geer et al., 2002*) [accessed on 29th October 2018]. The initial set comprised 52,953, 82, 204, and 131 sequences, respectively. To reduce the number of bacterial sequences to a computationally more tractable subset yet maintain sequence diversity, we pre-processed the bacterial sequence set as follows: first, each bacterial sequence was identified to family level using NCBI taxonomy annotations; then, only those sequences with a valid family-level taxonomic identification were retained (43,454 sequences belonging to 409 families). Within each family, we then calculated pairwise identities between all sequences and identified up to ten sequence identity clusters. Subsequently, a single representative sequence was randomly sampled from each cluster (for families comprising of fewer than 10 sequences, we selected all available sequences). This reduced the bacterial set to 3135 sequences.

Next, the full archaeal, eukaryotic, viral and reduced bacterial sets were processed using the Batch Web CD-Search Tool (*Marchler-Bauer and Bryant, 2004*) to determine the position and integrity of the HU-IHF domain(s). Based on the domain identification, we then selected only those sequences from each set that contained a single, complete domain. The bacterial set was further reduced by selecting only those sequences less than or equal to 110 amino acids. Given the relative scarcity of sequences in the other kingdoms, their sequences were not size-filtered. The final set comprised 30 archaeal, 164 eukaryotic, 112 viral, and 1920 bacterial sequences.

We note that the hits to archaeal (N = 30) and eukaryotic (N = 164) genomes that we obtained, arguably fall into two classes. The first class comprises putative homologs that are isolated from other archaeal/eukaryotic hits on the phylogeny in *Figure 2a*. In principle, these putative homologs might constitute rare cases of horizontal gene transfer (HGT), with a narrow phylogenetic footprint indicating recent arrival. In many, if not most instances, however, these cases likely represent bacterial contaminants in published genome assemblies. This particularly concerns a number of cases where purportedly eukaryotic/archaeal sequences branch with high support with a bacterial sequence. Some examples of such pairs are provided in *Supplementary file 3*. Further indicating likely contamination, several phylogenetically haphazard hits are found even in, for example, mammalian genomes, where rates of horizontal transfer are thought to be extremely low. A striking example is the genome of the Tibetan antelope (*Pantholops hodgsonii*), which – at face value – harbors three different HU proteins affiliated with divergent branches of the bacterial HU phylogeny

(*Figure 2a*, *Supplementary file 3*). Contamination might also be the most conservative (and parsimonious) explanation for some hits to phylogenetically isolated archaeal genomes, especially when these were assembled from metagenomic samples (*Figure 2a*, *Supplementary file 3*).

Hits belonging to the second class, in contrast, have at least some phylogenetic persistence and coherence, indicative of vertical inheritance. Amongst eukaryotes, this notably includes homologs in dinoflagellates, some algae, and apicomplexa (including Plasmodium, Theileria, and Babesia species) – all single-celled organism, which have either acquired HU proteins from their resident organelles or unrelated HGT events. As mentioned above, functional roles for HU proteins have been described for at least some species in these clades.

There are few putative HU homologs in archaeal genomes, with the majority of hits (23/30) belonging to the Thermoplasmatales/DHVE2 clade. Thermoplasmatales/DHVE2 hits are monophyletic (85% bootstrap support, *Figure 2b*), with the exception of a single sequence (Thermoplasmata M8B2D, PNX46291.1, *Figure 2a*), which clusters with 71% bootstrap support with HU from *Desulfobacula toluolica*, a proteobacterium, potentially indicative of contamination or recent HGT. A small number of sequences (4) from halophilic archaea, including *Halorubrum aidingense* JCM 13560 (EMA66313.1), are embedded amongst Bacteriodetes sequences and appear to form a reasonably coherent phylogenetic unit (*Figure 2—figure supplement 1*). Note, however, that the sequence search space contained many other genomes from halophilic archaea, including model organisms such as *Haloferax volcanii,* which did not yield any hits, suggesting that HU proteins in these four species might, again, represent recent acquisitions or assembly contaminants rather than established parts of the functional genome.

Regarding the likely origin of HTa in the Thermoplasmatales/DHVE2 clade, we note that relationships amongst deeper nodes are generally very poorly resolved (as illustrated by smaller node sizes in *Figure 2a*), including in relation to the Thermoplasmatales/DHVE2 clade. There is no strongly supported relationship with particular bacterial sequences that would suggest an ancestral bacterial donor clade. However, HGT remains the most parsimonious explanation for the phyletic distribution of hits, especially in light of the observation that HTa is absent from other members of the larger Diaforarchaea clade, within which Thermoplasmatales/DHVE2 branch.

Sequences were aligned using the Constraint-based Multiple Alignment Tool (COBALT) through the NCBI web-interface (https://www.ncbi.nlm.nih.gov/tools/cobalt/cobalt.cgi) with default parameters. The phylogenetic tree was reconstructed using RAxML (version 8.2.10) with the following parameters: -f a, -m PROTCATAUTO, -T 20. Branch support was based on 100 bootstrap calculations performed in RAxML (*Stamatakis, 2014*).

Scripts to replicate analyses presented here are available via https://github.com/hocherantoine/HTa_Histone_analog (copy archived at https://github.com/elifesciences-publications/HTa_Histone_analog).

# Acknowledgements

The authors thank Dennis Searcy for advice on *T. acidophilum* husbandry and chromatin biochemistry, Finn Werner for mentorship and his lab for feedback and advice, Harald Huber and the Archaeenzentrum in Regensburg for *M. fervidus* biomass, Jacques Oberto for *E. coli hupA/B* deletion strains, and Amy Schmid and Saaz Sakrikar for comments on the manuscript. This work was supported by a UKRI Innovation Fellowship (JBS), an Imperial College Integrative Experimental and Computational Biology Studentship (AE), and UK Medical Research Council core funding (TW).

# Additional information

## Funding

| Funder | Grant reference number | Author |
|---|---|---|
| Medical Research Council | MC_A658_5TY40 | Tobias Warnecke |

The funders had no role in study design, data collection and interpretation, or the decision to submit the work for publication.

## Author contributions

Antoine Hocher, Conceptualization, Data curation, Formal analysis, Validation, Investigation, Visualization, Methodology, Writing—original draft, Project administration, Writing—review and editing; Maria Rojec, Resources, Data curation, Investigation, Methodology; Jacob B Swadling, Formal analysis, Investigation, Visualization, Methodology; Alexander Esin, Formal analysis, Investigation, Methodology; Tobias Warnecke, Conceptualization, Formal analysis, Supervision, Funding acquisition, Investigation, Visualization, Methodology, Writing—original draft, Project administration, Writing—review and editing

## Author ORCIDs

Tobias Warnecke (iD) https://orcid.org/0000-0002-4936-5428

## Decision letter and Author response

Decision letter https://doi.org/10.7554/eLife.52542.SA1
Author response https://doi.org/10.7554/eLife.52542.SA2

# Additional files

## Supplementary files

• Supplementary file 1. Representation of HU homologs across bacterial phyla.

• Supplementary file 2. Examples of putative archaeal and eukaryotic homologs that likely represent contamination during genome assembly.

• Supplementary file 3. Fourier filtering parameters.

• Transparent reporting form

## Data availability

All sequencing data generated for this study have been deposited in GEO under accession code GSE127728.Scripts to replicate analyses are available via https://github.com/hocherantoine/HTa_Histone_analog (copy archived at https://github.com/elifesciences-publications/HTa_Histone_analog).

The following dataset was generated:

| Author(s) | Year | Dataset title | Dataset URL | Database and Identifier |
|---|---|---|---|---|
| Hocher A, Warnecke T | 2019 | The DNA-binding protein HTa from Thermoplasma acidophilum is an archaeal histone analog | https://www.ncbi.nlm.nih.gov/geo/query/acc.cgi?acc=GSE127728 | NCBI Gene Expression Omnibus, GSE127728 |

The following previously published datasets were used:

| Author(s) | Year | Dataset title | Dataset URL | Database and Identifier |
|---|---|---|---|---|
| Maruyama H, Harwood JC, Moore KM, Paszkiewicz K, Durley SC, Fukushima H, Atomi H, Takeyasu K, Kent NA | 2013 | Thermococcus kodakarensis MNase-Seq | https://www.ncbi.nlm.nih.gov/sra/?term=SRR495445 | NCBI Sequence Read Archive, SRR495445 |
| Ammar R, Torti D, Tsui K, Gebbia M, Durbic T, Bader GD, Giaever G, Nislow C | 2012 | Haloferax volcanii nucleosome map | https://www.ncbi.nlm.nih.gov/sra/?term=SRR574592 | NCBI Sequence Read Archive, SRR574592 |
| Rojec M, Warnecke T | 2019 | The role of archaeal histones in gene expression - a synthetic biology perspective | https://www.ncbi.nlm.nih.gov/geo/query/acc.cgi?acc=GSE127678 | NCBI Gene Expression Omnibus, GSE127678 |

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
