## [Decision Letter]

**Acceptance summary:**

This manuscript focuses on the HTa protein of *Thermoplasma acidophilum*, a hyperthermophilic archaeal species. HTa is a homologue of the bacterial HU protein; HU has been investigated extensively for its role in bacterial genome organization and gene regulation. Unlike many other archaeal species, *T. acidophilum* lacks histones. The authors show that HTa functions to package *T. acidophilum* DNA into a chromatin-like structure. Thus, the paper provides strong evidence that HTa functions equivalently to an archaeal histone. This is an important discovery because it shows there is more than one evolutionary path to packaging DNA into chromatin, and it explains how some archaea manage without histones. It will be exciting to see how HTa impacts *T. acidophilum* physiology, and whether other archaeal species lacking histones have evolved alternative ways to package their DNA.

**Decision letter after peer review:**

[Editors’ note: a previous version of this study was rejected after peer review, but the authors submitted for reconsideration. The first decision letter after peer review is shown below.]

Thank you for submitting your work entitled "The DNA-binding protein HTa from *Thermoplasma acidophilum* is an archaeal histone analog" for consideration by *eLife*. Your article has been reviewed by two peer reviewers, and the evaluation has been overseen by a Reviewing Editor and a Senior Editor. The following individuals involved in review of your submission have agreed to reveal their identity: Amy Schmid (Reviewer #2).

Our decision has been reached after consultation between the reviewers. Based on these discussions and the individual reviews below, we regret to inform you that your work will not be considered further for publication in *eLife*. The two major concerns of the reviewers, which I share, are (i) insufficient evidence that the genome-wide pattern of MNase-protected fragments in *T. acidophilum* is due to HTa, and (ii) uncertainty as to whether the HTa sequence specificity inferred from the MNase/sequencing datasets are due to HTa binding, or occur as a result of MNase biases. Both of these points are addressable with additional experiments. For example, point (i) could be addressed by analyzing the MNase-protected fragments from *E. coli* expressing HTa or immunoprecipitating HTa prior to MNase treatment, and point (ii) could be addressed by making libraries from genomic DNA digested with MNase and comparing to the MNase-protected DNA from living cells. Given the length of time it would take to perform these experiments, the manuscript has been rejected. However, if you were to do these experiments, and the overall conclusions of the study remained the same, we would encourage you to resubmit.

Reviewer #1:

Most archaeal species, but not all, encode histone proteins. Although the utility and requirements for histones differ between species, interest remains in how non-histone encoding archaeal species organize their genomes. A plethora of small basic proteins have been identified, primarily in the Crenarchaeaota, which impact genomic architectures. The manuscript in review approaches an outstanding question of whether the apparently horizontally transferred HU-like proteins found in some archaeal genomes plays a dominant role in chromosome organization. Unfortunately, the manuscript fails to resolve impactful questions of archaeal biology, chromosome organization, or the biological role of HTa to an extent that could warrant publication.

Hocher et al. present a detailed statistical analysis of MNase-seq data from the archaeon *T. acidophilum* to propose that a HU-like protein (HTa) binds the genome and organizes such in an manner analogous to the organization resultant from histone-binding in histone-encoding archaeal species. The results presented suggest that HTa may play a role in genome organization but no evidence for transcriptional regulation is supported by the current data. Through extensive mathematically and statistical manipulations, the authors largely come to the conclusions supported by work in the late 70s and early 80s… HTa is abundant, binds most of the genome, and likely can bind at adjacent positions or multimerize to generate larger complexes that result in larger regions of protection.

All of the conclusions and interpretations are from MNase-seq data, with no supporting in vitro data with purified HTa to support sequence-binding preferences or oligomerization potential. This is a major flaw. Genetic systems are not available for *T. acidophilum*, thus no biologically manipulations are possible and no inferences from the MNase-seq data can be borne-out at the organismal level. Although outside of the scope of what could be requested, the absence of biologically relevant data is disappointing. The results are thus likely of limited utility to the archaeal, chromatin, or transcription communities. Some of the mathematical modeling and statistical analyses could be of value to the community, but as applied here, offer little to no insight into biological function. Throughout the text, the authors repeatedly state that the existing data are not sufficient to support interpretations of biological function.

The current experimental work is not sufficient to support interpretations beyond broad interpretations that have been in the literature for ~40 years. While technically sound, the manuscript lacks impact and will not alter the course of future experimentation, inferences concerning archaeal evolution, genomic architecture, or transcription regulation in the third domain.

Reviewer #2:

Hocher et al. investigate the chromatin structure of *Thermoplasma acidophilum*, a hyperthermophilic archaeal species. In particular, they provide evidence that HTa, a homolog of bacterial HU, functions as a major chromatin protein of this species. The paper is built around MNase digestion and a thorough analysis of MNase-Seq data, including comparisons to other species. The experiments and analyses are well-designed, reporting the genome-wide positioning of chromatin DNA, its sequence preferences, possible oligomerisation of the HU molecular, as well as differences with its known bacterial homologs and convergence with archaeal histones. The data provides strong evidence that HTa acts as an archaeal histone analog, the central hypothesis of the paper. The work is exciting and novel from several standpoints. First, the study examines the chromatin structure of an archaeon that lacks a histone homolog, which has not yet been attempted. Second, and more importantly, the paper adds significantly to a small body of evidence that supports a new hypothesis that archaeal chromatin proteins are diverse, dynamic, and vary by phylogenetic lineage in terms of structure and function. There are a few points where further clarity on the analysis and text revisions are needed to support the main conclusions.

1) Was MNase-seq performed on *E. coli* heterologously expressing HTa? If so, this would help confirm which signals observed in the native host are specific to HTa and which require other proteins (e.g. those signals attributed to nucleation-dependent vs independent fragments). If not, please explain why.

2) Figure 4 I,J. This analysis reveals that protected segments are GC-rich. In particular, the analysis about the possible oligomerization: "nucleation" by a single molecule and extension to form a larger chromatin structure, mediated by GC-richness and hindered by ATrich sequences, is very thorough. However, as the authors note in subsection “HTa exhibits histone-like sequence preferences”, it has been reported that MNase preferentially digests DNA at AT-rich sequences, and hence the GC-richness of the apparent peaks could partly result from enzyme bias. In addition, the paper the authors cite regarding this issue (Allan et al., 2012) reports an insignificant difference in binding occupancy based on MNase vs CAD digestion, but to my knowledge, does not report on differences in sequence bias of regions protected during digestion by each of these enzymes. In contrast, (Chung et al., 2010) suggest using a matched control with MNase-digested naked DNA to correct for the bias. The authors of the current study mention a sonicated naked DNA control to correct for "…bias in coverage associated with replication…", but it is unclear whether the MNase-digested naked DNA control was performed as well. In support of the claims made, the authors do note "read-internal" sequence enrichments and AT-rich regions within protected fragments, neither of which would be expected from MNase bias alone. However, the sequence patterns of protected regions is a major conclusion of the paper. Given the data on hand, is there any way to correct for MNase digestion bias across the sequence analyses? If not, can the authors explain more clearly and specifically explain how the sequence bias of MNase could impact their results, for example differential digestion (complete vs incomplete) as an alternative cause/ interpretation of the different peak widths, and as an alternative explanation for the AT boundaries for small fragments, etc?

3) Figure 5. The authors examine nucleotide frequencies within the dyad. Interestingly, they appear to find a pattern of repeating AT or GC dinucleotides. It has been suggested (Struhl and Segal Nat Struct Mol Biol 2014) that a 10bp periodic repeat of such dinucleotides could facilitate histone-DNA interaction. Did the authors detect any periodicity in the observed MNase sequences? Across the genome as a whole?

[Editors’ note: what now follows is the decision letter after the authors submitted for further consideration.]

Thank you for resubmitting your work entitled "The DNA-binding protein HTa from *Thermoplasma acidophilum* is an archaeal histone analog" for further consideration by *eLife*. Your revised article has been evaluated by Kevin Struhl, Senior Editor and Joe Wade, Reviewing Editor.

The reviewers and editors agree that the new data provide the additional evidence needed to support the main conclusion of the paper: that HTa functions as a histone-like protein in *Thermoplasma acidophilum*. Consequently, I am happy to recommend the paper for publication with only a couple of minor additions, detailed below:

1) The GitHub repository given in the rebuttal document (https://github.com/hocherantoine) seems currently to be empty. It is important to populate this repository prior to publication so that others can reproduce the results.

2) For Figure 6E and F, it would be helpful to provide more information. Specifically, how many reads are represented by each of the data points (the size of the circles only shows a relative number – what is the absolute number?), and are the differences between P_slow_ scores significant?

---

## [Author Response]

[Editors’ note: the author responses to the first round of peer review follow.]

Our decision has been reached after consultation between the reviewers. Based on these discussions and the individual reviews below, we regret to inform you that your work will not be considered further for publication in eLife. The two major concerns of the reviewers, which I share, are (i) insufficient evidence that the genome-wide pattern of MNase-protected fragments in T. acidophilum is due to HTa, and (ii) uncertainty as to whether the HTa sequence specificity inferred from the MNase/sequencing datasets are due to HTa binding, or occur as a result of MNase biases. Both of these points are addressable with additional experiments. For example, point (i) could be addressed by analyzing the MNase-protected fragments from E. coli expressing HTa or immunoprecipitating HTa prior to MNase treatment, and point (ii) could be addressed by making libraries from genomic DNA digested with MNase and comparing to the MNase-protected DNA from living cells. Given the length of time it would take to perform these experiments, the manuscript has been rejected. However, if you were to do these experiments, and the overall conclusions of the study remained the same, we would encourage you to resubmit.

We have carried out the experiments suggested by the reviewers. First, to address point (i), we have analysed MNase-Seq data from *E. coli* expressing HTa. We find qualitatively identical nucleotide enrichment patterns in the fragments retrieved from HTa-expressing *E. coli* but not in reads from an *E. coli* genomic DNA digest (Figure 5F in the manuscript). This finding, discussed in subsection “HTa exhibits histone-like sequence preferences” of the revised manuscript, supports the notion that the patterns observed in *T. acidophilum* chiefly reflect HTa binding.

To address point (ii), we made libraries from MNase-digested *T. acidophilum* naked genomic DNA. These libraries exhibit the typical MNase cleavage signature at the end of reads, demonstrating that MNase worked as expected. Importantly, however, the correlation in coverage between native chromatin and genomic DNA is poor (ρ=0.07, Figure 5G) and we find no nucleotide enrichment patterns inside reads from our genomic DNA library (Figure 5F). These results, also discussed in subsection “HTa exhibits histone-like sequence preferences” of the revised manuscript, argue strongly against the possibility that the native chromatin digest is to any great extent a reflection of MNase digestion biases.

We wanted to go further than the experiments suggested by the reviewers and establish beyond reasonable doubt that HTa exhibits a preference for GC-rich sequences and that, as we suggested in the initial version of the manuscript, intrinsic sequence preferences are a key driver of HTa binding in vivo. We therefore carried out two experiments not explicitly requested by the reviewers but which we thought would best allay lingering doubts (discussed in subsection “HTa exhibits histone-like sequence preferences” of the revised manuscript).

First, we reconstituted HTa and *T. acidophilum* genomic DNA in vitro, digested the samples with MNase, and sequenced the resulting fragments. We find an excellent correlation between HTa occupancy in vitro and in vivo (small fragments: ρ=0.62, P< 2.2e-16) and a LASSO model trained on the in vitro data predicts the observed in vivo data remarkably well (new Figure 6B, ρ=0.68, P< 2.2e-16), strongly supporting the view that sequence is the principal determinant of HTa occupancy in vivo.

Second, to provide orthogonal evidence for sequence preferences that does not involve MNase digestion, we designed and carried out an EMSA-seq experiment. Briefly, based on the *T. acidophilum* in vivo digest, we selected five 100bp regions across the *T. acidophilum* genome with very low to very high (5th, 25th, 50th, 75th, 95th percentile) occupancy. We then generated a large and diverse library of double-stranded DNA oligos with different GC contents by introducing random dinucleotides at 8 defined positions along these five backbones. Combining this library with purified HTa, we then carried out electrophoretic mobility shift assays (EMSAs), which revealed distinct bands corresponding to differential HTa binding (new Figure 6C,D). Analyzing differential composition of oligos across bands migrating at different speed, we find a marked enrichment for GC-rich sequences in the slow-migrating HTa-bound band (Figure 6E,F and figure supplements).

Thus, MNase digestion of HTa-expressing *E. coli, T. acidophilum* genomic DNA, in vitro reconstituted *T. acidophilum* chromatin and orthogonal EMSA experiments all support and further bolster our original conclusions, that HTa prefers GC-rich sequences, positions according to intrinsic sequence preferences in vivo and therefore exhibits key characteristics associated with histones in other organisms.

Reviewer #1:Most archaeal species, but not all, encode histone proteins. Although the utility and requirements for histones differ between species, interest remains in how non-histone encoding archaeal species organize their genomes. A plethora of small basic proteins have been identified, primarily in the Crenarchaeaota, that impact genomic architectures. The manuscript in review approaches an outstanding question of whether the apparently horizontally transferred HU-like proteins found in some archaeal genomes plays a dominant role in chromosome organization. Unfortunately, the manuscript fails to resolve impactful questions of archaeal biology, chromosome organization, or the biological role of HTa to an extent that could warrant publication.

We did not set out to answer whether HTa plays dominant role in chromatin organization – Searcy’s work had already shown that HTa is the principal constituent of *T. acidophilum* chromatin. Rather, in charting the binding footprints of HTa genome-wide – and thereby providing the first global chromatin landscape in an archaeon without histones – we discovered something unexpected. HTa, unlike its bacterial orthologs, does not prefer AT-biased sequences or lacks sequence preferences altogether. Instead, it favours GC-rich sequences in a fashion remarkably reminiscent of histones. As we see it, the main impact of this manuscript does not lie in providing a detailed mechanistic understanding of the functional role(s) of HTa in *T. acidophilum*, but in the demonstration that a protein of bacterial origin has converged onto properties normally associated with histone proteins, with broad implications for our understanding of chromatin evolution and evolvability. To name but one: our results show that the presence of different principal building blocks need not imply different chromatin architecture but that, in fact, two proteins with fundamentally different folds can give rise to very similar types of organization.

We agree with the reviewer that much remains to be done in dissecting the biological roles of HTa. However, while our work lays strong foundations for future investigations in this area, its principal merits, as highlighted above, lie elsewhere.

Hocher et al. present a detailed statistical analysis of MNase-seq data from the archaeon T. acidophilum to propose that a HU-like protein (HTa) binds the genome and organizes such in an manner analogous to the organization resultant from histone-binding in histone-encoding archaeal species. The results presented suggest that HTa may play a role in genome organization but no evidence for transcriptional regulation is supported by the current data. Through extensive mathematically and statistical manipulations, the authors largely come to the conclusions supported by work in the late 70s and early 80s… HTa is abundant, binds most of the genome, and likely can bind at adjacent positions or multimerize to generate larger complexes that result in larger regions of protection.

We think it is reassuring that we have come to similar conclusions regarding HTa abundance, multimerization behaviour, and capacity to protect from MNase digestion. We claim no novelty here, but rather provide corroboration/replication and additional refinement and quantification on which subsequent novel results are built.

All of the conclusions and interpretations are from MNase-seq data, with no supporting in vitro data with purified HTa to support sequence-binding preferences or oligomerization potential. This is a major flaw.

The reviewer is right to point out that in vitroexperiments would be desirable to confirm intrinsic binding preferences and oligomerization potential. As this is a key point of the manuscript, we embarked on a set of additional experiments: a) reconstituting purified HTa with *T. acidophilum* genomic DNA, and b) carrying out an EMSA-seq experiment. As described in response to the editorial summary, these orthogonal in vitroresults further strengthen the conclusions drawn from in vivoMNase-Seq data.

Genetic systems are not available for T. acidophilum, thus no biologically manipulations are possible and no inferences from the MNase-seq data can be borne-out at the organismal level. Although outside of the scope of what could be requested, the absence of biologically relevant data is disappointing. The results are thus likely of limited utility to the archaeal, chromatin, or transcription communities. Some of the mathematical modeling and statistical analyses could be of value to the community, but as applied here, offer little to no insight into biological function. Throughout the text, the authors repeatedly state that the existing data are not sufficient to support interpretations of biological function.

As highlighted above, we did not set out to dissect the biological functions of HTa. In the absence of a genetic system, this is a difficult proposition – as the reviewer is keenly aware. We therefore think it honest and scholarly to highlight the fact that current data offer little insight into biological function. Since we did generate steady state transcriptomic data, we thought it interesting to ask whether there are correlations between HTa binding and transcriptional change. We did not see any such correlations. Whether this is because HTa has genuinely no dynamic role in regulating transcription will, as we point out, require further investigation.

The current experimental work is not sufficient to support interpretations beyond broad interpretations that have been in the literature for ~40 years. While technically sound, the manuscript lacks impact and will not alter the course of future experimentation, inferences concerning archaeal evolution, genomic architecture, or transcription regulation in the third domain.

We thank the reviewer for attesting to the technical soundness of the research. With regard to the perceived impact of our research on the understanding of archaeal chromatin biology and evolution, please see our response to point #1.

Reviewer #2:Hocher et al. investigate the chromatin structure of Thermoplasma acidophilum, a hyperthermophilic archaeal species. In particular, they provide evidence that HTa, a homolog of bacterial HU, functions as a major chromatin protein of this species. The paper is built around MNase digestion and a thorough analysis of MNase-Seq data, including comparisons to other species. The experiments and analyses are well-designed, reporting the genome-wide positioning of chromatin DNA, its sequence preferences, possible oligomerisation of the HU molecular, as well as differences with its known bacterial homologs and convergence with archaeal histones. The data provides strong evidence that HTa acts as an archaeal histone analog, the central hypothesis of the paper. The work is exciting and novel from several standpoints. First, the study examines the chromatin structure of an archaeon that lacks a histone homolog, which has not yet been attempted. Second, and more importantly, the paper adds significantly to a small body of evidence that supports a new hypothesis that archaeal chromatin proteins are diverse, dynamic, and vary by phylogenetic lineage in terms of structure and function. There are a few points where further clarity on the analysis and text revisions are needed to support the main conclusions.

We thank the reviewer for her enthusiastic appraisal of our manuscript.

1) Was MNase-seq performed on E. coli heterologously expressing HTa? If so, this would help confirm which signals observed in the native host are specific to HTa and which require other proteins (e.g. those signals attributed to nucleation-dependent vs independent fragments). If not, please explain why.

We have now performed MNase-Seq on HTa-expressing *E. coli* as well as *E. coli* naked genomic DNA. As discussed in the response to the editor’s summary, these experiments (as well as *E. coli* native chromatin digest data reported we recently reported elsewhere) support our previous conclusions, showing, as they do, that digestion fragments from HTa-expressing *E. coli* but not naked DNA bear the same signature nucleotide enrichments we found in *T. acidophilum* native chromatin.

2) Figure 4 I,J. This analysis reveals that protected segments are GC-rich. In particular, the analysis about the possible oligomerization: "nucleation" by a single molecule and extension to form a larger chromatin structure, mediated by GC-richness and hindered by ATrich sequences, is very thorough. However, as the authors note in subsection “HTa exhibits histone-like sequence preferences”, it has been reported that MNase preferentially digests DNA at AT-rich sequences, and hence the GC-richness of the apparent peaks could partly result from enzyme bias. In addition, the paper the authors cite regarding this issue (Allan et al., 2012) reports an insignificant difference in binding occupancy based on MNase vs CAD digestion, but to my knowledge, does not report on differences in sequence bias of regions protected during digestion by each of these enzymes. In contrast, (Chung et al., 2010) suggest using a matched control with MNase-digested naked DNA to correct for the bias. The authors of the current study mention a sonicated naked DNA control to correct for "…bias in coverage associated with replication…", but it is unclear whether the MNase-digested naked DNA control was performed as well. In support of the claims made, the authors do note "read-internal" sequence enrichments and AT-rich regions within protected fragments, neither of which would be expected from MNase bias alone. However, the sequence patterns of protected regions is a major conclusion of the paper. Given the data on hand, is there any way to correct for MNase digestion bias across the sequence analyses? If not, can the authors explain more clearly and specifically explain how the sequence bias of MNase could impact their results, for example differential digestion (complete vs incomplete) as an alternative cause/ interpretation of the different peak widths, and as an alternative explanation for the AT boundaries for small fragments, etc?

We have now sequenced and analyzed MNase-digested *T. acidophilum* genomic (naked) DNA and compared it to the native chromatin digest. We note the following: a. considering genome-wide occupancy, the correlation between naked DNA and chromatin digests is poor (ρ=0.07, new Figure 5G), inconsistent with MNase biases as the major driver of the sequence signals we observe. b. In line with this, while cutting biases at the edge of fragments are expectedly still observed, we find no read-internal enrichment patterns that are present in the chromatin digest (new Figure 5F). c. Explicitly normalizing observed in vivooccupancy by read coverage derived from genomic DNA digests does not affect conclusions (something we do not show in the revised manuscript to avoid redundancy but mention here for the sake of completeness).

3) Figure 5. The authors examine nucleotide frequencies within the dyad. Interestingly, they appear to find a pattern of repeating AT or GC dinucleotides. It has been suggested (Struhl and Segal Nat Struct Mol Biol 2014) that a 10bp periodic repeat of such dinucleotides could facilitate histone-DNA interaction. Did the authors detect any periodicity in the observed MNase sequences? Across the genome as a whole?

We do not find a periodic 10bp dinucleotide repeat pattern comparable to eukaryotes in *T. acidophilum*. MNase fragments display symmetric/asymmetric enrichments (Figure 5A-C), but we see no evidence for periodicity akin to what is seen in eukaryotes or even *T. kodakarensis* (Figure 5A-C). There is also no significant ~10bp periodicity in the *T. acidophilum* genome as a whole (Author response image 1) as determined using PerScan (Mrazek et al. 2011 Microb Inform Exp 1:13). Note that periods at ~11-12bp are usually attributed to supercoiling state/DNA helical pitch.

**Author response image 1. respfig1:** Nucleotide periodicities in the *T. acidophilum* genome.

[Editors' note: the author responses to the re-review follow.]

The reviewers and editors agree that the new data provide the additional evidence needed to support the main conclusion of the paper: that HTa functions as a histone-like protein in Thermoplasma acidophilum. Consequently, I am happy to recommend the paper for publication with only a couple of minor additions, detailed below:1) The GitHub repository given in the rebuttal document (https://github.com/hocherantoine) seems currently to be empty. It is important to populate this repository prior to publication so that others can reproduce the results.

The GitHub repository (https://github.com/hocherantoine/HTa_Histone_analog) is now populated with the relevant scripts to reproduce the analyses in the paper.

2) For Figure 6E and F, it would be helpful to provide more information. Specifically, how many reads are represented by each of the data points (the size of the circles only shows a relative number – what is the absolute number?), and are the differences between P_slow_ scores significant?

We have added information on absolute read counts to Figure 6E/F. In the figure, we now also provide correlation coefficients (weighted Pearson correlation) between GC (GpC) content and P_slow_. As you might have anticipated, the relationships are highly significant (weighted r>0.85, P<0.001).